# A systematic review and meta-analysis on international studies of prevalence, mortality and survival due to coal mine dust lung disease

Cynthia Lu[1], Paramita Dasgupta[1], Jessica Cameron[1,2], Lin Fritschi[3], Peter Baade[1,2,4]*

1 Cancer Council Queensland, Brisbane, Queensland, Australia, 2 School of Mathematical Sciences, Queensland University of Technology, Gardens Point, Brisbane, Queensland, Australia, 3 School of Public Health, Curtin University, Bentley, Western Australia, Australia, 4 Menzies Health Institute Queensland, Griffith University, Gold Coast Campus, Southport, Queensland, Australia

☯ These authors contributed equally to this work.
* peterbaade@cancerqld.org.au

## Abstract

### Background

Coal mine dust lung disease comprises a group of occupational lung diseases including coal workers pneumoconiosis. In many countries, there is a lack of robust prevalence estimates for these diseases. Our objective was to perform a systematic review and meta-analysis of published contemporary estimates on prevalence, mortality, and survival for coal mine dust lung disease worldwide.

### Methods

Systematic searches of PubMed, EMBASE and Web of Science databases for English language peer-reviewed articles published from 1/1/2000 to 30/03/2021 that presented quantitative estimates of prevalence, mortality, or survival for coal mine dust lung disease. Review was conducted per PRISMA guidelines. Articles were screened independently by two authors. Studies were critically assessed using Joanna Briggs Institute tools. Pooled prevalence estimates were obtained using random effects meta-analysis models. Heterogeneity was measured using the $I^2$ statistics and publication bias using Egger's tests.

### Results

Overall 40 studies were included, (31 prevalence, 8 mortality, 1 survival). Of the prevalence estimates, fifteen (12 from the United States) were retained for the meta-analysis. The overall pooled prevalence estimate for coal workers pneumoconiosis among underground miners was 3.7% (95% CI 3.0–4.5%) with high heterogeneity between studies. The pooled estimate of coal workers pneumoconiosis prevalence in the United States was higher in the 2000s than in the 1990s, consistent with published reports of increasing prevalence following decades of declining trends. Sub-group analyses also indicated higher prevalence

**Data Availability Statement:** All the studies included in this systematic review are already published. All relevant data from the published

articles are within the manuscript and its
Supporting information Files.

**Funding:** This study was supported by funding
from Resources Safety and Health Queensland
(Prevalence study of coal mine dust lung disease
Contract OHHFY20-2). PB received the funding.
The funders did not play any role in study design,
data collection, analysis and interpretation of data,
decision to submit for publication or preparation of
the manuscript. https://www.rshq.qld.gov.au/.

**Competing interests:** The authors have declared
that no competing interests exist.

among underground miners, and in Central Appalachia. The mortality studies were suggestive of reduced pneumoconiosis mortality rates over time, relative to the general population.

## Conclusion

The ongoing prevalence of occupational lung diseases among contemporary coal miners highlights the importance of respiratory surveillance and preventive efforts through effective dust control measures. Limited prevalence studies from countries other than the United States limits our understanding of the current disease burden in other coal-producing countries.

## Introduction

Coal mine dust lung disease (CMDLD) comprises a group of occupational lung diseases caused by the cumulative inhalation of respirable coal mine dust particles including carbon, quartz, and silicates [1, 2]. These diseases include coal workers pneumoconiosis (CWP, commonly referred to as black lung disease), mixed dust pneumoconiosis, silicosis, dust-related diffuse fibrosis and chronic obstructive pulmonary disease (COPD). The spectrum of CMDLD also includes progressive massive fibrosis (PMF), the most severe form of CWP.

While there are no curative treatments [3], CWP and other CMDLD can be prevented by reducing exposure to coal dust through technological advances [1, 4] or enforcing statutory dust standards by regulatory means [2, 3]. Engineering controls to limit respirable mine dust exposure include improved ventilation and dust suppression techniques such as water sprays [4, 5]. In many high-income countries, government guidelines also specify occupational exposure limits for respirable coal dust, however these can vary between countries [2, 3]. For example, in the United States (USA) the occupational exposure limit for coal dust is 1.5 mg/m$^3$ and crystalline silica is 0.05 mg/m$^3$ while corresponding limits for Australia were 3.0 mg/m$^3$ and 0.1 mg/m$^3$ respectively until September 2020 [2] when they were revised to the USA levels [5]. Despite modern dust controls being a primary preventative measure, there is currently limited evidence on the performance and effectiveness of these regulatory controls [1, 2, 6].

Although the prevalence of CMDLD decreased markedly from the 1970s to 1990s, recent studies have reported an unexpected increase in CWP cases [1, 2, 6]. The increase in CWP prevalence has occurred even among miners who were subject to regulatory guidelines their entire working life [7]. Factors contributing to this pattern may include having mines with a smaller workforce, longer working hours and higher coal mine dust levels caused by more mechanised mining technologies [1, 2, 6]. Consequently, there have been renewed public health concerns about these debilitating and progressive diseases [1], highlighting the importance of better understanding the disease burden and prevention in the modern era.

In many countries, including Australia, there is a lack of prevalence estimates for occupational lung diseases, primarily because of the lack of national comprehensive mandatory reporting systems [3]. This limits the ability to quantify the historical trends and current disease burden and to direct policy and prevention to the most appropriate areas. In an attempt to quantify historical and current burden of disease, in this study, we systematically reviewed published international estimates of prevalence, mortality, and survival for CMDLD and derived summary measures through meta-analyses.

## Methods

### Protocol

This review was conducted according to the Preferred Reporting Items for Systematic Reviews and Meta Analyses (PRISMA) guidelines (S1 Checklist) [8]. The clinical question to guide the review was defined following the structured PICOS framework. The PICOS question for this study was: among coal mine workers (Participants), under current coal dust control strategies (Intervention), from the start to the end of the study time period (Comparison) what was the estimated prevalence or mortality or survival (Outcome) for CMDLD using cohort studies (Study design)?

A protocol for this systematic review and meta-analysis was not registered or published prior to this work commencing and has not been subsequently submitted since protocols cannot be registered retrospectively.

### Disease definitions

CMDLD can be divided into two main groups: fibrotic or nodular disease (includes CWP) and non-nodular disease such as COPD [1]. Confirmation of a CMDLD diagnosis involves compatible history and medical imaging through chest radiographs and high-resolution computed tomography screening [1].

For surveillance purposes, included prevalence studies in this review defined CWP cases based on the International Labour Office (ILO) International Classification of Radiographs of Pneumoconiosis [9] which classifies chest abnormalities (opacities) by size (small opacities are lesions up to 1 cm and large opacities are those >1 cm), shape, and profusion. Small opacity profusion is defined using four categories of increasing profusion (0, 1, 2, 3). Each of the four categories is further divided into three sub-categories resulting in a 12-point scale (from (0/- to 3/+), where the first number indicates the category finally assigned to a profusion and the second a seriously considered alternative. Silicosis is defined by the presence of profusion category $\geq 1/0$ and r-type opacities which are rounded small opacities (0.3 to 1.0 cm). CWP is defined as presence of small opacities with ILO profusion sub-category $\geq 1/0$, PMF as the presence of any large opacity, and advanced CWP as a profusion sub-category of $\geq 2/1$. If there is development of PMF and/or an increase in small opacity profusion greater than one ILO sub-category over five years, this is indicative of rapidly progressive CWP.

### Search strategy

The electronic databases PUBMED and EMBASE were systematically searched for all indexed articles from 1 January 2000 onwards to focus on more contemporary estimates. The Web of Science database was used for cited reference searches. Searches were last performed on 15 February 2021. Search strategies used selected subject headings and key words related to coal mine workers, for example, "coal mining", "coal miners", "surface coal miners", "underground coal miners", combined with terms pertaining to CMDLD, including "coal worker's pneumoconiosis", "anthracosis", "silicosis", "dust related diffuse fibrosis", "progressive massive fibrosis", "chronic obstructive pulmonary disease", and outcome measures of "prevalence, mortality or survival" (S1 Table). Reference lists of published review papers were checked to identify additional potentially relevant articles.

### Study selection

Studies were eligible if they met the following inclusion criteria:

1. Study cohort included active and/or former coal mine workers and/or sub-groups; and

2. Outcome measure was CMDLD prevalence, mortality, or survival among coal miners, and

3. Quantitative estimates of CMDLD prevalence, mortality or survival were presented, or for prevalence only, could be calculated from the reported information.

The scope of the review was limited to English-language peer-reviewed original research articles. Reviews, editorials, commentaries, and conference abstracts were excluded. However, the reference lists of all reviews identified through queries and articles that met inclusion criteria were searched for additional studies.

The titles and abstracts of all unique articles identified by the searches were independently reviewed for eligibility by two reviewers (CL, PD). Discrepancies were discussed and resolved through consensus. The full text versions of all potentially relevant articles were then assessed by one reviewer (CL) and categorised as "include" or "exclude" with reasons for exclusion being documented. A second reviewer (PD) blinded to the primary reviewer's (CL) decisions checked the article selection for the final review. These decisions were compared and discussed and if necessary, the senior author (PB) was also consulted.

## Data extraction

For all included articles, one reviewer (CL) extracted data on source (first author, year and title), study characteristics (including data source, location, time period, design), population (such as size, mine type), disease definition, number of cases, study methods and statistical findings for outcome measures relevant to the review question and objectives. For prevalence studies information on duration of estimates was also collected. A second reviewer (PD) independently checked the data extract against the original source for all included articles.

Multiple studies from the same geographical area using the same data source were screened to avoid including duplicate data in the subsequent meta-analysis. The region covered by three states within the USA, Kentucky, Virginia, and West Virginia, is referred to throughout this review as 'Central Appalachia'.

For each type of disease (Table 1), studies were sorted alphabetically by author within country for each outcome measure. For CWP, prevalence estimates are first presented for all cases combined and then separately by population sub-groups (mining tenure, mine size, mine type, geographical location) and disease severity (PMF, rapidly progressive or advanced CWP). The term 'mining tenure' in the context of this review refers to the duration of employment in the mining industry. Some studies are repeated either in the same table or across multiple tables.

**Table 1. The spectrum of coal mine dust lung disease (CMDLD).**

| Lung pathological changes | Diagnoses |
|---|---|
| Fibrotic or nodular | Coal workers' pneumoconiosis (CWP)<br>• Simple CWP<br>• Rapidly progressive/advanced CWP<br>• Progressive massive fibrosis (PMF) |
|  | Silicosis |
|  | Mixed dust pneumoconiosis |
|  | Diffuse dust-related fibrosis |
| Non-nodular | Chronic obstructive pulmonary disease (COPD)<br>• Chronic bronchitis<br>• Emphysema |
| Mixed | Fibrotic CMDLD with COPD |

## Critical appraisal

Included prevalence studies were assessed for risk of bias using the Joanna Briggs Institute (JBI) critical appraisal checklist for studies reporting prevalence data [10, 11] as recommended by a recent review on critical appraisal tools for various study designs [12]. This checklist contains nine questions that assess specific domains to determine the potential risk of bias. These domains look at the sampling approach (2 questions), representativeness of the study population (1 question), sample transferability and credibility (2 questions), participation rate (1 question), use of valid measures for case definition and measurement (2 questions) and statistical approach (1 question).

The JBI critical appraisal checklist for cohort studies [10] was used for the mortality/survival studies included in this review. This included 11 domains pertaining to selection bias (1 question), exposure measurement (2 questions), confounding factors (2 questions), outcome assessment (2 questions), follow-up time (3 questions) and statistical approach (1 question). For each study, relevant questions were answered with 'yes', 'no', 'unclear' or 'not applicable'.

The overall risk of bias of individual studies was then determined with the following cut-offs: low risk of bias if at least 70% of all answers were yes (prevalence: $7 \leq score \leq 9$; cohort: $8 \leq score \leq 11$), moderate risk if 50 to 69% questions were yes (prevalence: $5 \leq score \leq 6$; cohort: $6 \leq score \leq 7$) and high risk of bias if yes answers were below 50% (prevalence: $0 \leq score \leq 4$; cohort $0 \leq score \leq 5$) [13, 14]. These tools are included in S2 Table.

The risk of bias evaluation was used to help evaluate the quality of evidence from each study. However, to be as comprehensive as possible, all studies meeting the inclusion criteria were included in the review, regardless of their quality score. It is also worth noting that even if there is a very large bias in one domain (such as sampling), it can only impact the bias score by either ±2 (prevalence) or ±1 (mortality/survival). This means that while some prevalence estimates, for example those using the federal black lung benefits program database [7, 15], are biased upward compared to the total coal mine population, this has only a minor impact on the total bias score for these studies.

## Statistical analysis

We used random-effects meta-analysis models [16] that allow for the inherent heterogeneity of observational studies to pool published prevalence estimates by CMDLD type. Only studies that reported both population size and number of disease cases and were deemed to be sufficiently homogenous in terms of study characteristics including geographical location and sampled population were retained for the meta-analysis. This determination was subjective, however it meant, for example, that studies based on worker compensation data or other non-surveillance-based data sources were not combined with those based on population-based surveillance of coal miners. To avoid duplication, only the estimate from the most recently published study was used if two studies used the same data source for the same disease over identical time periods. However, if the excluded study also reported prevalence for another CMDLD and met the other criteria it was retained for the corresponding meta-analysis.

Pooled estimates for CWP prevalence were stratified by disease severity. Additional analysis by sub-groups such as coal mine type, country, geographical location (USA) and study time period were only carried out if there were at least two studies per strata. As such, the analyses stratified by time period (1990s compared with 2000s) were only possible for CWP and PMF in the USA, with the choice of these two periods determined by the included studies.

Additional sub-group analyses by region were also conducted to generate summary estimates of CWP prevalence in the USA only in the most recent time period (2005–2009).

Sensitivity analyses were carried out to explore the impact of retaining only the four high quality (low risk of bias) studies on the pooled prevalence estimates for CWP among underground coal miners in the USA. Sub-group analysis by risk-of-bias for other disease types was not possible due to the limited number of studies.

Summary estimates for mortality were not generated because of wide heterogeneity in the reported measures and/or lack of required information for pooling estimates across studies. There was only one study presenting survival results.

Heterogeneity between study-specific estimates was assessed with the Q and $I^2$ statistics [17] with $I^2$ values of 25%, 50% and 75% being the cut-offs for indicating low, moderate and high heterogeneity respectively. In the initial exploratory meta-analysis, we looked at the impact of excluding/including specific studies on the $I^2$ measure of between-study heterogeneity. We used the Egger's test [18] to assess publication bias. Sensitivity analyses assessed the influence of individual studies on the pooled estimates by repeating the meta-analysis omitting one study at a time.

Meta-analysis was performed with Stata/SE version 16 (StataCorp, TX, USA). Forest plots were used for graphical presentation of results. Summary estimates of prevalence are reported with 95% confidence intervals (CI).

### Presentation of results

Overall study findings are summarized in a narrative form to supplement the results of the meta-analysis and to synthesize data from studies not included in the meta-analysis. Included tables provide contextual information for the textual commentary.

## Results

### Study selection

As shown in a PRISMA diagram (Fig 1), a total of 405 articles were identified through the search queries with two more identified through other sources (that is references identified through the reference lists of review articles). The removal of duplicates left 225 potentially eligible records. After initial screening of the title and abstracts, 106 were excluded. Of the remaining 119 full-text articles that were evaluated for eligibility, 40 were retained for the review (31 prevalence, 8 mortality and 1 survival), of which 15 were eligible for meta-analyses (15 prevalence, 0 mortality and 0 survival). Of the excluded 79 (full text) articles, 46 mentioned prevalence/mortality in the abstract but did not specifically report quantitative estimates of these measures, while 25 were review articles (or commentaries) that did not report original estimates. Of the remaining eight articles, two were not available in English, and a full-text version could not be found for six other articles.

### Study characteristics

Detailed characteristics of the individual studies are summarized in Table 2. Around three-quarters (71%) of the 40 included studies in this review were from the USA (n = 29) [7, 15, 19–45] followed by China (n = 4) [46–49] and the United Kingdom (UK, n = 3) [50–52]. One study each was from South Africa [53]; Turkey [54], Ukraine [55], and Australia [56]. All studies either used the ILO classification system [9] or one based on it (China) to define CWP, silicosis and r-type opacities. Nineteen of the 31 cross-sectional prevalence studies were sourced from surveillance databases while remaining articles either accessed other administrative collections or involved surveys. Patterns of prevalence for CMDLD by key disease types and

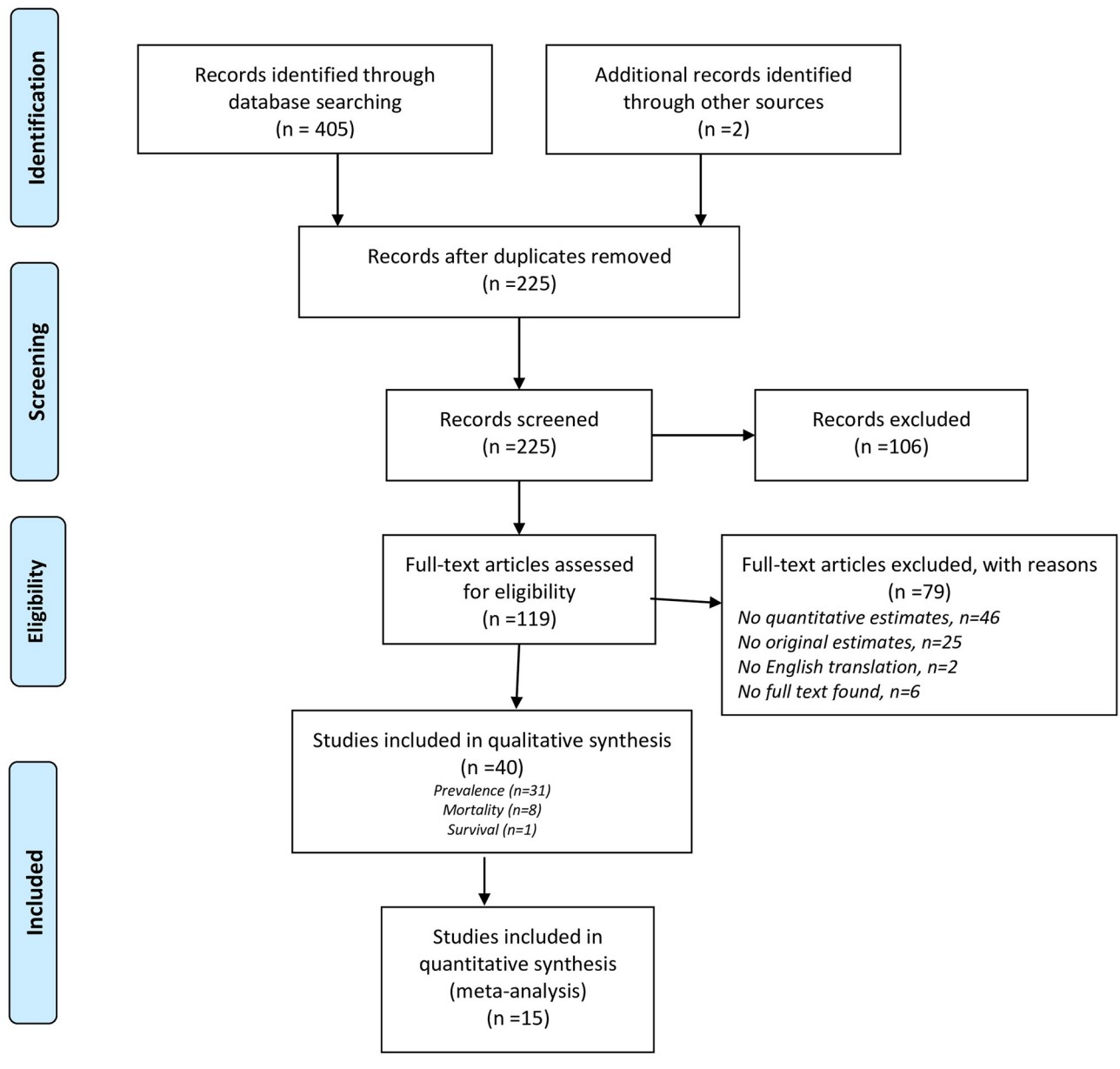

**Fig 1. PRISMA flowchart for study selection.**

country across the 31 included studies in this systematic review are shown graphically in S1 Fig.

All nine articles on mortality or survival were population-based observational cohort studies using official mortality records.

The study population for the majority of the 31 prevalence studies comprised underground miners only (n = 22), three studies included surface miners only while remaining studies were not restricted to a specific mine type. Only two studies presented prevalence estimates for CWP further stratified by mine size, while two others only reported these estimates for long-term miners (at least 20 years) and four more only for Central Appalachian region of the USA.

**Table 2. Summary of included studies in systematic review.**

| Author, year | Country | Period | Data source | Coal mine type | CMDLD[a] | Population | Score | Category |
|---|---|---|---|---|---|---|---|---|
| | | | | | | | Risk of bias [b,c] | |
| Studies arranged alphabetically by first author by country for each outcome measure | | | | | | | | |
| **Prevalence** | | | | | | | | |
| Almberg et al. 2018 [15] | USA | 1970–2016 | Compensation[d] | All | PMF | 341,176 | 6 | Moderate |
| Antao et al. 2005 [19] | USA | 1996–2002 | Surveillance[e] | Underground | CWP, rapidly progressive CWP | 29,521 | 7 | Low |
| Arif et al. 2020 [20] | USA | 2011–2014 | Medicare[f] | Underground | CWP | 541,262 | 7 | Low |
| Blackley et al. 2014 [21] | USA | 2005–2012 | Surveillance[g] | Underground | CWP, PMF, lung function abnormality | 3,771 | 7 | Low |
| Blackley et al. 2018a [22] | USA | 1970–2017 | Surveillance[h] | Underground (only ≥20 years tenure) | CWP | NS | 6 | Moderate |
| Blackley et al. 2018b [23] | USA | 2013–2017 | Clinical records[i] | Underground | PMF, r-type opacities | 11,200 | 6 | Moderate |
| CDC 2000 [24] | USA | 1996–1997 | Survey | Surface | Silicosis | 1,236 | 6 | Moderate |
| CDC 2003 [25] | USA | 1996–2002 | Surveillance[e] | Underground and surface | CWP, PMF | 31,179 | 7 | Low |
| CDC 2006 [26] | USA | 2006 | Surveillance[g] | Underground | advanced CWP, PMF | 328 | 6 | Moderate |
| CDC 2007 [27] | USA | 2006 | Surveillance[g] | Underground | advanced CWP | 975 | 6 | Moderate |
| CDC 2012 [28] | USA | 2010–2011 | Surveillance[h] | Surface | CWP, PMF, advanced CWP, r-type opacities | 2,257 | 6 | Moderate |
| Graber et al. 2017 [7] | USA | 2001–2013 | Compensation[d] | Underground | CWP, PMF, advanced CWP | 24,686 | 5 | Moderate |
| Hall et al. 2019 [29] | USA | 1980–2018 | Surveillance[h] | Underground | r-type opacities | 106,506 | 7 | Low |
| Hall et al. 2020 [30] | USA | 2014–2019 | Surveillance[h] | Surface | CWP | 6,790 | 6 | Moderate |
| Kurth et al. 2020 [31] | USA | 2005–2016 | Surveillance[g] | Underground and surface | CWP, PMF | 5,316 | 6 | Moderate |
| Laney & Atfield 2010 [32] | USA | 1970–2009 | Surveillance[g,h] | Underground | CWP, PMF | 145,512 | 7 | Low |
| Laney & Atfield 2014 [33] | USA | 1970–2009 | Surveillance[h] | Underground (only ≥25 years tenure) | CWP | NS | 6 | Moderate |
| Laney et al. 2010 [34] | USA | 1980–2008 | Surveillance[j] | Underground | CWP, PMF, rapidly progressive CWP, r-type opacities | 90,973 | 7 | Low |
| Laney et al. 2012 [35] | USA | 2005–2009 | Surveillance[g] | Underground | CWP, PMF, advanced CWP, r-type opacities | 6.658 | 6 | Moderate |
| Laney et al. 2017 [36] | USA | 2000–2016 | Surveillance[h] | Underground | PMF | 60,205 | 7 | Low |
| Reynolds et al. 2017 [37] | USA | 2005–2015 | Surveillance[g] | Underground and surface | CWP, PMF, lung function abnormality | 5,605 | 6 | Moderate |
| Suarthana et al. 2011 [38] | USA | 2005–2009 | Surveillance[h] | Underground | CWP | 12,408 | 7 | Low |
| Wang et al. 2013 [39] | USA | 2005–2009 | Surveillance[g] | Underground | CWP, PMF, lung function abnormality | 6,373 | 6 | Moderate |
| Vallyathan et al. 2011 [40] | USA | 1971–1996 | Autopsy[k] | Underground | PMF | NS | 4 | High |
| Han et al. 2015 [46] | China | 1963–2014 | Survey | Underground | CWP | 8,928 | 4 | High |
| Han et al. 2016 [47] | China | 1965–2012 | Survey | All | CWP | 19,116 | 6 | Moderate |
| Li et al. 2018 [48] | China | 1960–2004 | Survey | Opencast | CWP | 8,191 | 4 | High |
| Scarisbrick et al. 2002 [50] | UK | 1998–2000 | Surveillance[l] | All | CWP | 4,647 | 6 | Moderate |
| Naidoo et al. 2004 [53] | South Africa | NS | Survey | Underground | CWP | 896 | 6 | Moderate |
| Tor et al. 2010 [54] | Turkey | 1985–2004 | Survey | Underground | CWP | NS | 4 | High |
| Graber et al. 2012 [55] | Ukraine | 2000–2002 | Survey | Underground | COPD | 1,065 | 6 | Moderate |
| **Mortality/survival** | | | | | | | | |
| Attfield & Kuempel 2008. [41] | USA | 1969/71–1993 | Death registry[m] | All | Pneumoconiosis (CWP, silicosis), COPD | 8,899 | 8 | Low |

*(Continued)*

**Table 2.** (Continued)

| Author, year | Country | Period | Data source | Coal mine type | CMDLD[a] | Population | Risk of bias [b,c] Score | Category |
|---|---|---|---|---|---|---|---|---|
| Beggs et al. 2015 [42] | USA | 2003–2013 | Death registry[n] | All | Pneumoconiosis, CWP | NS | 8 | Low |
| Bell & Mazurek. 2020 [43] | USA | 1998–2018 | Death registry[n] | All | CWP | NS | 8 | Low |
| CDC 2010 [44] | USA | 1968–2006 | Death registry[n] | All | CWP | NS | 8 | Low |
| Graber et al 2014. [45] | USA | 1969/71–2007 | Death registry[o] | All | Pneumoconiosis (CWP, silicosis), COPD | 9,033 | 9 | Low |
| Coggon et al. 2010 [51] | UK | 1979–2000 | Death registry[p] | All | CWP, silicosis, COPD | NS | 7 | Moderate |
| Miller & MacCalman. 2010 [52] | UK | 1959–2005 | Death registry[p] | All | CWP, silicosis, COPD | NS | 8 | Low |
| Smith et al. 2006 [56] | Australia | 1979–2002 | Death registry[q] | All | CWP, pneumoconiosis due to silica dust | NS | 8 | Low |
| Han et al. 2017 [49] | China | 1963–2014 | Hospital records | All | CWP | 452 | 5 | High |

CDC, Centre for Disease Control and Prevention; CMDLD, coal mine dust lung disease; COPD, chronic obstructive bronchitis; CWP, coal workers pneumoconiosis; NS, not stated; PMF, progressive massive fibrosis; UK, United Kingdom; USA, United States

[a] Cases of pneumoconiosis, silicosis and r-type opacities defined based on International Labour Office classification system [9], lung function abnormalities by spirometry and COPD (clinical diagnosis).

[b] Risk of bias based on Joanna Briggs Institute critical appraisal tools for prevalence or cohort (mortality) studies. Please refer to text for further details.

[c] Risk of bias ranked as low risk of bias if at least 70% of all answers were yes (prevalence: 7≤score≤9; cohort: 8≤ score ≤11), moderate risk if 50 to 69% questions were yes (prevalence: 5≤score≤6; cohort:6 ≤ score ≤7) and high risk of bias if yes answers were below 50% (prevalence: 0≤score≤4; cohort 0 ≤ score ≤5).

[d] Federal black lung program benefits

[e] Coal Workers' X-ray Surveillance Program (CWXSP) and Miners' Choice Program (MCP).

[f] The 5% Medicare limited claims dataset

[g] Enhanced Coal Workers' Health Surveillance Program CWHSP (ECWHSP)

[h] Coal Workers' Health Surveillance Program (CWHSP)

[i] Three federally funded black lung clinics

[j] Coal Workers' X-ray Surveillance Program (CWXSP)

[k] The USA national coal workers' autopsy study

[l] Periodic X-ray scheme

[m] Social Security administration death data, Internal Revenue service records, Department of Vital Statistics

[n] National Centre for Health Statistics' multiple cause of death files

[o] National Death Index

[p] Office of National Statistics

[q] Australian Bureau of Statistics national mortality data set

The cohort size ranged from 6,658 to 541,262 with a mean of 49,617. Calendar years covered by the different studies also varied widely and ranged between 1959 and 2019. Twelve studies reported long-term CMDLD prevalence (at least 15 years), 12 reported short-term (≤ 5 years) estimates and seven presented prevalence estimates over six to 14 years. On average, prevalence estimates were reported over a 10-year study period and mortality estimates over a 25-year period.

## Outcome measures

Thirty-one studies either reported prevalence estimates (percentage of CMDLD cases among cohort) or provided sufficient information on number of cases and total cohort size so that

prevalence estimates could be calculated. Twenty-eight studies reported the prevalence of CWP and/or at least one CWP subtype by disease severity, while three studies reported prevalence estimates for another disease in the CMDLD spectrum. Thirteen studies looked at more than one type of CMDLD.

One study each also described trends in prevalence for PMF and chronic bronchitis in terms of the annual percentage change, however no study reported linear trends in CWP prevalence over time.

Included mortality studies used various outcome measures such as standardised mortality ratio, age standardised mortality rate and excess deaths per year.

## Risk of bias

For 31 prevalence studies, the overall risk of bias score ranged from four to seven out of a total of nine domains in the JBI critical appraisal tool for prevalence studies (S3a Table). Nine studies were deemed to have low risk of bias, 18 moderate and remaining four high risk of bias (Table 2). A key limiting factor for 22 studies from the USA was related to sampling and selection bias in the underlying data source. Participation in the national surveillance programs was voluntary and limited to current coal miners, while compensation databases only included coal miners applying for corresponding benefits. This means that the cohorts in these USA studies may not be completely representative of all active or former miners who are at risk of being diagnosed. For those studies categorised as having moderate risk of bias, a key limitation was the relatively small or not stated sample size. Three studies at high risk of bias used cross-sectional surveys but lacked information on response rate and/or were unable to demonstrate that the sample was representative of the population. The remaining study was based on autopsy data and most of the appraisal questions were considered not applicable or uncertain. Across all 31 prevalence studies, CMDLD cases were defined consistently and reliably using a standard measure, such as the ILO classification system [9].

Seven of eight mortality studies were considered to have a low risk of bias with scores of eight to nine out of 11 domains for JBI cohort tool (S3b Table), with the remaining one classified as moderate (score 7/11) (Table 2). While most did not identify or adjust for confounders, follow-up was sufficiently long and complete and all studies were based on an objective and reliable population-based data source (mortality records). One survival study was classified as high risk of bias (score = 5/11, S3b Table), having unclear information about follow-up, a non-representative study sample and ambiguous statistical analysis.

## Overall prevalence of CWP

Of 17 included studies that reported overall CWP prevalence (Table 3), 11 were from the USA, three from China and one each from the UK, Turkey, and South Africa. There was considerable variability in the prevalence of CWP by country with estimates ranging from 0.8% to 6.2% in China [46–48]; the UK [50], South Africa [53] and Turkey [54]. Several studies based on national surveillance data of the current coal miner workforce in the USA reported estimates of 2.1–5.1% [19, 25, 31, 32, 34, 35, 37–39] although they varied in their time periods and duration (years). Two other studies used compensation data to report widely varying estimates of 0.2% among Medicare beneficiaries [20] and 33.8% among miners who applied for benefits through the federal black lung benefits program [7], probably reflecting differences in data collection methods.

**Table 3. Summary of included studies on overall prevalence of coal workers pneumoconiosis.**

| Author, year[a] | Country | Period | Data source | Cases[b] | Population | Prevalence duration (years) | Prevalence (%) |
|---|---|---|---|---|---|---|---|
| Antao et al. 2005 [19] | USA | 1996–2002 | CWXSP, MCP | 886 | 29,521 | 7 | 3.0 |
| *Arif et al. 2020 [20]* | *USA* | *2011–2014* | *Medicare[c]* | *1,021* | *541,262* | *4* | *0.2* |
| *CDC 2003 [25]* | *USA[d]* | *1996–2002* | *CWXSP, MCP* | *862* | *31,179* | *7* | *2.8* |
| *Graber et al. 2017 [7]* | *USA* | *2001–2013* | *Federal BLPB* | *8,355* | *24,686* | *13* | *33.8* |
| *Kurth et al. 2020 [31]* | *USA[e]* | *2005–2016* | *ECWHSP* | *116* | *5,316* | *12* | *2.2* |
| Laney & Atfield 2010 [32] | USA | 1970–2009 | CWHSP, ECWHSP | 7,276 | 145,512 | 40 | 5.0 |
| Laney et al. 2010 [34] | USA | 1980–2008 | CWXSP | 2,868 | 90,973 | 29 | 3.2 |
| *Laney et al. 2012 [35]* | *USA[f]* | *2005–2009* | *ECWHSP* | *276* | *6,658* | *5* | *4.1* |
| *Reynolds et al. 2017 [37]* | *USA[g]* | *2005–2015* | *ECWHSP* | *103* | *4,985* | *11* | *2.1* |
| Suarthana et al. 2011 [38] | USA | 2005–2009 | CWHSP | 446 | 12,408 | 5 | 3.6 |
| Wang et al. 2013 [39] | USA | 2005–2009 | ECWHSP | 255 | 6,373 | 5 | 4.0 |
| Han et al. 2015 [46] | China | 1965–2012 | Survey | 495 | 8,928 | 48 | 5.5 |
| Han et al. 2016 [47] | China | 1963–2014 | Survey | 411 | 19,116 | 52 | 2.2 |
| Li et al. 2018 [48] | China | 1960–2004 | Survey | 259 | 8,191 | 45 | 3.2 |
| *Scarisbrick et al. 2002 [50]* | *UK* | *1998–2000* | *PXR scheme* | *35* | *4,647* | *3* | *0.8* |
| *Naidoo et al. 2004 [53]* | *South Africa[h]* | *NS* | *Survey* | *NS* | *896* | *NS* | *2.0–4.0* |
| *Tor et al. 2010 [54]* | *Turkey[h]* | *1985–2004* | *Survey* | *NS* | *NS* | *20* | *1.2–6.2* |

BLPB, Black Lung Program Benefits; CDC, Centre for Disease Control and Prevention; CWHSP, Coal Workers' Health Surveillance Program; CWXSP, Coal Workers' X-ray Surveillance Program; ECWHSP, Enhanced Coal Workers' Health Surveillance Program; MCP, Miners' Choice Program; NS, not stated; PXR, Periodic X-ray; USA, United States

[a] Studies in italics not included in meta-analysis

[b] Based on International Labour Office classification system [9] coal workers pneumoconiosis was defined as presence of small opacities with profusion subcategory $\geq$ 1/0 on chest radiographs.

[c] The 5% Medicare limited claims dataset

[d] Not included in meta-analysis as data source and prevalence duration same as study by Antao *et al.* 2005 [19]

[e] Only non-smoking miners

[f] Not included in meta-analysis as data source and prevalence duration same as study by Wang *et al* 2013 [39]

[g] Excludes states of Kentucky, Virginia, and West Virginia

[h] Not included in meta-analysis as cohort and/or case size not stated.

## Prevalence estimates by CWP disease severity

**Progressive massive fibrosis (PMF).** The estimated prevalence of PMF (the most severe CWP subtype) ranged from 0.2 to 7.1% across 15 included studies, all from the USA (Table 4) [7, 15, 21, 23, 25, 26, 28, 31, 32, 34–37, 39, 40]. Magnitude of the estimates varied by data source, study period, and coal mining region. For example, higher PMF prevalence was reported for miners from Central Appalachian region than among coal miners nationwide based on surveillance data [21, 26].

Higher prevalence estimates were reported among past and current coal miners who were claimants to the federal black lung benefits program [7, 15] than among those coal miners included in the national surveillance databases [31, 32, 34–37, 39]. This is to be expected, since the black lung benefits cohort only included those coal miners who had made a compensation claim for a debilitating lung disease.

**Advanced/rapidly progressive CWP.** The prevalence of advanced forms of CWP in the USA was also higher among Central Appalachian miners [26–28] than other regions or nationally [28, 35]. It was also higher among the federal black lung program cohort [7] than studies using national surveillance data [19, 34] probably reflecting differences in population, as mentioned in preceding section (Progressive massive fibrosis).

**Table 4. Summary of included studies on prevalence of coal workers pneumoconiosis (CWP) by disease severity.**

| Author, year[a] | Country | Period | Data source | Cases | Population | Prevalence duration (years) | Prevalence (%) |
|---|---|---|---|---|---|---|---|
| | | | **Progressive massive fibrosis (PMF)[b]** | | | | |
| *Almberg et al. 2018* [15] | *USA* | *1970–2016* | *Federal BLPB* | *4,679* | *314,176* | *47* | *1.5* |
| *Blackley et al. 2014* [21] | *USA[c]* | *2005–2012* | *ECWHSP* | *53* | *3,771* | *8* | *1.4* |
| *Blackley et al. 2018b* [23] | *USA[d]* | *2013–2017* | *Clinical records* | *416* | *11,200* | *5* | *3.7* |
| CDC 2003 [25] | USA | 1996–2002 | CWXSP, MCP | 62 | 31,179 | 7 | 0.2 |
| *CDC 2006* [26] | *USA[d]* | *2006* | *ECWHSP* | *5* | *328* | *1* | *1.5* |
| *CDC 2012* [28] | *USA[e]* | *2010–2011* | *CWHSP* | *12* | *2,257* | *2* | *0.5* |
| Graber et al. 2017 [7] | USA | 2001–2013 | BLPB | 977 | 24,686 | 13 | 4.0 |
| *Kurth et al. 2020* [31] | *USA[f]* | *2005–2016* | *ECWHSP* | *31* | *5,316* | *12* | *0.6* |
| Laney & Atfield 2010 [32] | USA | 1970–2009 | CWHSP, ECWHSP | 485 | 145,512 | 40 | 0.3 |
| Laney et al. 2010 [34] | USA | 1980–2008 | CWXSP | 180 | 90,973 | 29 | 0.2 |
| *Laney et al. 2012* [35] | *USA[g]* | *2005–2009* | *ECWHSP* | *49* | *6,658* | *5* | *0.7* |
| Laney et al. 2017 [36] | USA | 2000–2016 | CWHSP | 225 | 60,205 | 17 | 0.4 |
| *Reynolds et al. 2017* [37] | *USA[h]* | *2005–2015* | *ECWHSP* | *10* | *4,985* | *11* | *0.2* |
| Wang et al. 2013 [39] | USA | 2005–2009 | ECWHSP | 45 | 6,373 | 5 | 0.7 |
| *Vallyathan et al. 2011* [40] | *USA[i]* | *1971–1996* | *NCWAS* | *NS* | *6,055* | *26* | *7.1* |
| | | | **Rapidly progressive CWP or advanced CWP[b]** | | | | |
| Antao et al. 2005 [19] | USA | 1996–2002 | CWXSP, MCP | 277 | 29,521 | 7 | 0.9 |
| *CDC 2006* [26] | *USA[d]* | *2006* | *ECWHSP* | *11* | *328* | *1* | *3.4* |
| CDC 2007 [27] | USA[j] | 2006 | ECWHSP | 37 | 975 | 1 | 3.8 |
| CDC 2012 [28] | USA[e] | 2010–2011 | CWHSP | 17 | 2,257 | 2 | 0.8 |
| | USA[k] | | | 11 | 833 | | 1.3 |
| | USA[h] | | | 6 | 1,424 | | 0.4 |
| Graber et al. 2017 [7] | USA | 2001–2013 | Federal BLPB | 1,110 | 24,686 | 13 | 4.5 |
| Laney et al. 2010 [34] | USA | 1980–2008 | CWXSP | 589 | 90,973 | 29 | 0.6 |
| Laney et al. 2012 [35] | USA | 2005–2009 | ECWHSP | 78 | 6,658 | 5 | 1.2 |

BLPB, Black Lung Program Benefits; CDC, Centre for Disease Control and Prevention; CWHSP, Coal Workers' Health Surveillance Program; CWXSP, Coal Workers' X-ray Surveillance Program; ECWHSP, Enhanced Coal Workers' Health Surveillance Program; MCP, Miners' Choice Program; NCWAS, National Coal Workers' Autopsy Study; NS, not stated; USA: United States

[a] Studies in italics not included in meta-analysis

[b] Based on International Labour Office classification system [9] PMF was defined as the presence of any large opacity (>1cm), rapidly progressive CWP as PMF development and/or an increase in small opacity profusion > one profusion subcategory over five years; and advanced CWP as profusion subcategory of ≥2/1 on chest radiographs.

[c] Not included in meta-analysis as estimates only for Central Appalachian region (states of Kentucky, Virginia, and West Virginia)

[d] Not included in meta-analysis as estimates only for state of Virginia

[e] Not included in meta-analysis as estimates only for surface coal miners

[f] Only non-smoking miners

[g] Not included in meta-analysis as data source and prevalence duration same as study by Wang *et al* 2013 [39]

[h] Excludes states of Kentucky, Virginia, and West Virginia

[i] Not included in meta-analysis as cohort size not stated.

[j] Only for states of Kentucky and Virginia

[k] Only for Central Appalachian region (states of Kentucky, Virginia, and West Virginia)

## Changes over time in CWP prevalence

The prevalence of CWP was also found to vary over time (S2 Fig) with one study from the USA reporting that after a downwards trend over 30 years from 1970s onwards, it started to increase again in the 2000s [32]. Similar temporal trends were also reported among

long-tenured miners in the USA, with this pattern being most evident among Central Appalachian miners [22, 33], however no quantitative estimates of the magnitude of the trend were presented. In the UK, CWP prevalence also decreased from the 1960s to early 1990s, then increased again [50]. Only one study reported trends in PMF prevalence and found it began increasing since 1978 in the USA with a sharper increase from 1998 to 2016 [15].

## Prevalence of CWP by population sub-groups

Key patterns in prevalence of CWP by population sub-groups are summarized below (Table 5).

**Coal mine type.** There was a suggestion that CWP prevalence was higher among bituminous underground coal miners than for surface miners, although differences in time periods and prevalent duration between studies made definitive statements difficult. For example, in the late 2000's the 5-year prevalence was estimated to be 10.1% among Central Appalachian bituminous underground coal miners [38], while another study reported an overall CWP prevalence of between 2.0–4.0% among bituminous underground miners in South Africa although the actual prevalence duration was not stated [53]. By contrast, the overall CWP prevalence among surface miners in the USA was around 2.0% nationally [25, 28, 30, 37], and about twice that in Central Appalachia [28, 30].

**Mining tenure.** Longer mining tenure was consistently associated with higher CWP prevalence [22, 25, 33, 38] among underground miners across all coal mining regions in the USA.

**Mine size.** Workers from larger mines (at least 50 employees) had a lower CWP prevalence than those from smaller mines in the USA both nationally [25] and in the Central Appalachian region [21].

**Geographical location (USA only).** Regional patterns were also observed in reported estimates for USA. Underground miners from Central Appalachia had consistently higher CWP prevalence in the late 2000's than underground miners from other mining regions in the USA [35, 38]. This same regional pattern was evident among both short [38] and long-term tenured miners [22, 38]. For example, among underground miners with at least 25 years tenure, CWP prevalence from 2013–2017 was 4-fold higher in Central Appalachia than other states combined [22].

**By country.** The estimated prevalence of CWP among coal miners ranged from 0.8% in the UK [50], 1.2%-6.2% in Turkey, China and South Africa [46–48, 53, 54] and 2.0–3.0% in several studies from the USA based on national surveillance data [19, 25, 31, 32, 34, 35, 37–39] over a similar time period.

## Prevalence estimates for other CMDLD

Five studies, all from the USA reported prevalence estimates for r-type opacities, a type of lung abnormality indicative of silicosis (Table 6). The long-term overall prevalence of r-type opacities among underground miners was stable over recent decades [29, 34]. Among underground miners in Central Appalachia, the five-year prevalence of r-type opacities in the late 2000's was almost twice the national prevalence [35] while the corresponding estimate solely for the Central Appalachian state of Virginia was very similar to the national prevalence [23]. Reported prevalence of these abnormalities among surface miners in the USA [28] was also very similar to the national prevalence among underground miners.

The prevalence of silicosis among surface miners from 1996–1997 in the USA was 6.7% [24] while COPD prevalence among underground miners in the Ukraine from 2000–2002 was 14.9% [55]. Finally, the prevalence of lung abnormalities among miners in the USA from 2005 to mid- 2010's over three studies ranged from 9.3% to 14.6% [21, 37, 39].

**Table 5. Summary of included studies on prevalence of coal workers pneumoconiosis by population sub-groups.**

| Author, year | Country | Period | Data source | Cases | Population | Prevalence duration (years) | Prevalence (%) |
|---|---|---|---|---|---|---|---|
| **Underground (Bituminous)** | | | | | | | |
| Suarthana et al. 2011 [38] | USA[b] | 2005–2009 | CWHSP | 294 | 2,914 | 5 | 10.1 |
| Naidoo et al. 2004 [53] | South Africa | NS | Survey | NS | 896 | NS | 2.0–4.0 |
| Tor et al. 2010 [54] | Turkey | 1985–2004 | Survey | NS | NS | 20 | 1.2–6.2 |
| **Surface mines** | | | | | | | |
| CDC 2003 [25] | USA | 1996–2002 | CWXSP, MCP | 196 | 10,466 | 7 | 1.9 |
| CDC 2012 [28] | USA | 2010–2011 | CWHSP | 46 | 2,257 | 2 | 2.0 |
| | USA[b] | | | 31 | 833 | | 3.7 |
| | USA[c] | | | 15 | 1,424 | | 1.1 |
| Hall et al. 2020 [30] | USA | 2014–2019 | CWHSP | 109 | 6,790 | 6 | 1.6 |
| | USA[b] | | | 44 | 935 | | 4.7 |
| | USA[c] | | | 61 | 5,855 | | 1.1 |
| *Reynolds et al. 2017* [37] | *USA[c]* | *2005–2015* | *ECWHSP* | *41* | *2,193* | *11* | *1.9* |
| **Mining tenure (years) >20** | | | | | | | |
| Blackley et al. 2018a [22] | USA | 2013–2017 | CWHSP (≥25 years) | NS | NS | 5 | 10.0 |
| | USA[b] | | | NS | NS | | 20.6 |
| | USA[c] | | | NS | NS | | 5.0 |
| CDC 2003 [25] | USA | 1996–2002 | CWXSP, MCP (≥25 years) | 367 | 6,778 | 7 | 5.4 |
| Laney & Atfield 2014 [33] | USA | 2005–2009 | CWHSP | NS | NS | 5 | 6.5 |
| | USA[b] | | | NS | NS | | 8.6 |
| Suarthana et al. 2011 [38] | USA[b] | 2005–2009 | CWHSP | NS | NS | 5 | 14.9 |
| | USA[c] | | | | | | 3.4 |
| **Short Mining tenure (years) ≤25** | | | | | | | |
| CDC 2003 [25] | USA | 1996–2002 | CWXSP, MCP | 11,610 | 280 | 7 | 2.4 |
| Suarthana et al. 2011 [38] | USA[b] | 2005–2009 | CWHSP (≤20 years) | NS | NS | 5 | 2.7 |
| | USA[c] | | | NS | NS | | 0.6 |
| **Mine size: small: ≤50 employees** | | | | | | | |
| Blackley et al. 2014 [21] | USA[b] | 2005–2012 | ECWHSP | 98 | 908 | 8 | 10.8 |
| CDC 2003 [25] | USA | 1996–2002 | CWXSP, MCP | NS | NS | 7 | 5.6 |
| **Mine size: large: >50 employees** | | | | | | | |
| Blackley et al. 2014 [21] | USA[b] | 2005–2012 | ECWHSP | 148 | 2,863 | 8 | 5.2 |
| CDC 2003 [25] | USA | 1996–2002 | CWXSP, MCP | NS | NS | 7 | 2.0 |
| **Geographical location (USA)** | | | | | | | |
| Blackley et al. 2014 [21] | USA[b] | 2005–2012 | ECWHSP | 246 | 3,771 | 8 | 6.5 |
| Laney et al. 2012 [35] | USA[b] | 2005–2009 | ECWHSP | 226 | 3,521 | 5 | 6.4 |
| | USA[c] | | | 50 | 3,137 | | 1.6 |
| Suarthana et al. 2011 [38] | USA[b] | 2005–2009 | CWHSP | 294 | 2,914 | 5 | 10.1 |
| | USA[c] | | | 152 | 9,494 | | 1.6 |

CDC, Centre for Disease Control and Prevention; CWHSP, Coal Workers' Health Surveillance Program; CWXSP, Coal Workers' X-ray Surveillance Program;

ECWHSP, Enhanced Coal Workers' Health Surveillance Program; MCP, Miners' Choice Program; NS, not stated; USA, United States

[a] Coal workers pneumoconiosis defined based on International Labour Office classification system [9].

[b] Only for Central Appalachian region (states of Kentucky, Virginia, and West Virginia)

[c] Excludes states of Kentucky, Virginia, and West Virginia

## Meta-analysis for CMDLD prevalence

Fifteen (48%) of 31 prevalence studies were retained for the meta-analysis. Excluded studies typically did not report overall estimates (n = 7) [22, 23, 26, 27, 31, 33, 37], or did not present

**Table 6. Summary of included studies on prevalence of other coal mine dust lung disease.**

| Author, year[a] | Disease[b] | Country | Data source | Period | Cases | Population | Prevalence duration (years) | Prevalence (%) |
|---|---|---|---|---|---|---|---|---|
| *Blackley et al. 2018b* [23] | *r-type opacities* | *USA*[c] | *Clinical records* | *2013–2017* | *122* | *11,200* | *5* | *1.1* |
| *CDC 2012* [28] | *r-type opacities* | *USA*[d] | *CWHSP* | *2010–2011* | *17* | *2,257* | *2* | *0.8* |
| Hall et al. 2019 [29] | r-type opacities | USA | CWHSP | 1980–2018 | 532 | 106,506 | 39 | 0.5 |
| Laney et al. 2010 [34] | r-type opacities | USA | CWXSP | 1980–2008 | 321 | 90,973 | 29 | 0.4 |
| Laney et al. 2012 [35] | r-type opacities | USA | ECWHSP | 2005–2009 | 83 | 6,658 | 5 | 1.2 |
| | | USA[e] | ECWHSP | 2005–2009 | 74 | 3,521 | 5 | 2.1 |
| CDC 2000 [24] | Silicosis | USA (PA)[d] | Survey | 1996–1997 | 83 | 1,236 | 2 | 6.7 |
| Graber et al. 2012 [55] | COPD | Ukraine | Survey | 2000–2002 | 159 | 1,065 | 2 | 14.9 |
| Blackley et al. 2014 [21] | Lung function abnormality | USA[e] | ECWHSP | 2005–2012 | 551 | 3,771 | 8 | 14.6 |
| Reynolds et al. 2017 [37] | Lung function abnormality | USA[f] | ECWHSP | 2005–2015 | 524 | 5,605 | 11 | 9.3 |
| Wang et al. 2013 [39] | Lung function abnormality | USA | ECWHSP | 2005–2009 | 836 | 6,373 | 5 | 13.1 |

CDC, Centre for Disease Control and Prevention; COPD, chronic obstructive bronchitis; CWHSP, Coal Workers' Health Surveillance Program; CWXSP, Coal Workers' X-ray Surveillance Program; ECWHSP, Enhanced Coal Workers' Health Surveillance Program; KY, Kentucky; PA, Pennsylvania; VA, Virginia; WV, West Virginia; USA, United States

[a] Studies in italics not included in meta-analysis

[b] Based on International Labour Office classification system [9] silicosis was defined as the presence of profusion category ≥1/0 and r-type opacities which are associated with silicosis as rounded radiographic opacities (3 to 10 mm). Lung function abnormalities were diagnosed based on abnormal spirometry results and COPD diagnosed based on clinical symptoms of chronic bronchitis and associated conditions.

[c] Not included in meta-analysis as estimates only for state of Virginia

[d] Not included in meta-analysis as estimates only for surface coal miners

[e] Only for Central Appalachian region (states of Kentucky, Virginia, and West Virginia)

[f] Excludes states of Kentucky, Virginia, and West Virginia

sufficient information to be included (n = 3) [40, 53, 54]. Three studies based on compensation databases [7, 15, 20] were also dropped as denominator was not comparable to surveillance-based studies. In addition, the single study from the UK [50] and one study each that reported estimates for only COPD [55] or silicosis [24] were excluded. Although two other studies reported results for CWP [25, 35] and one study for PMF [35] prevalence, they were not included for those specific meta-analyses to avoid potential duplication with other studies. However, these three studies met the criteria for inclusion in meta-analysis for the prevalence of other types of CMDLD.

Overall pooled estimate for CWP prevalence (Fig 2) over eight studies was 3.7% (95% CI 2.9–4.5%) with a high and significant heterogeneity between studies ($I^2$ = 98.9%, P <0.001). Sub-group analysis by country indicated that summary point estimates were very similar for USA (3.7% (95% CI 3.0–4.5%)) over five studies and China (3.6% (95% CI 1.6–5.6%)) over three studies, although confidence intervals were wide. Sensitivity analysis indicated that these estimates were not sensitive to individual studies (S3 Fig).

Analyses of CWP subtypes showed that the prevalence estimates of PMF in the USA was 0.3% (95% CI 0.2–0.5%) over five studies (Fig 3) and 0.9% (95% CI 0.6–1.2%) for advanced CWP over three studies (Fig 4). For both these diseases, inter-study heterogeneity was high (PMF: $I^2$ = 97.8%, advanced CWP $I^2$ = 94.8%) and statistically significant (P <0.001). Finally, the pooled prevalence of r-type opacities over three studies, all from the USA was 0.7% (95% CI 0.2–1.2%) with a high and significant heterogeneity ($I^2$ = 96.8%, P <0.001) (Fig 5). Sensitivity analysis indicated that these estimates were not sensitive to individual studies (S4 Table).

Summary estimates of CWP prevalence among surface miners in the USA across three studies was 1.8% (95% CI 1.6–2.0%) with low and non-significant heterogeneity ($I^2$ = 24.8%,

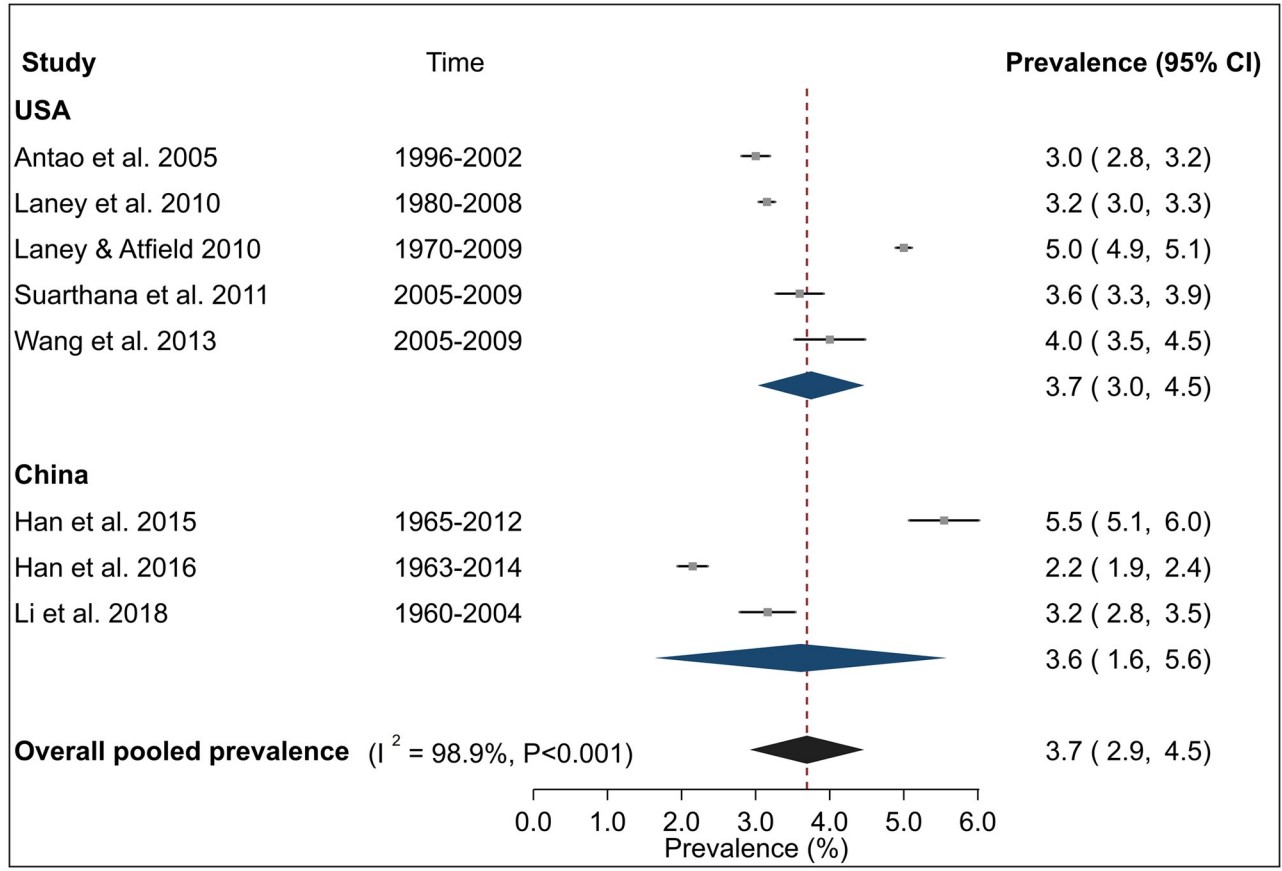

**Fig 2. Forest plot from meta-analysis for overall prevalence of coal worker's pneumoconiosis (CWP) among underground coal miners.**
Abbreviations are USA United States. The red dashed line corresponds to the overall effect size to guide the eye.

P = 0.248) (Fig 6) between studies. By contrast, the pooled CWP prevalence nationally among underground miners in the USA (including Central Appalachia) across five studies was 3.7% (95% CI 3.0–4.5%), among Central Appalachian miners across two studies was 8.3% (95% CI 4.8–11.8% and among miners in the rest of the USA was 1.6% (95% CI 1.4–1.8%) (Fig 7).

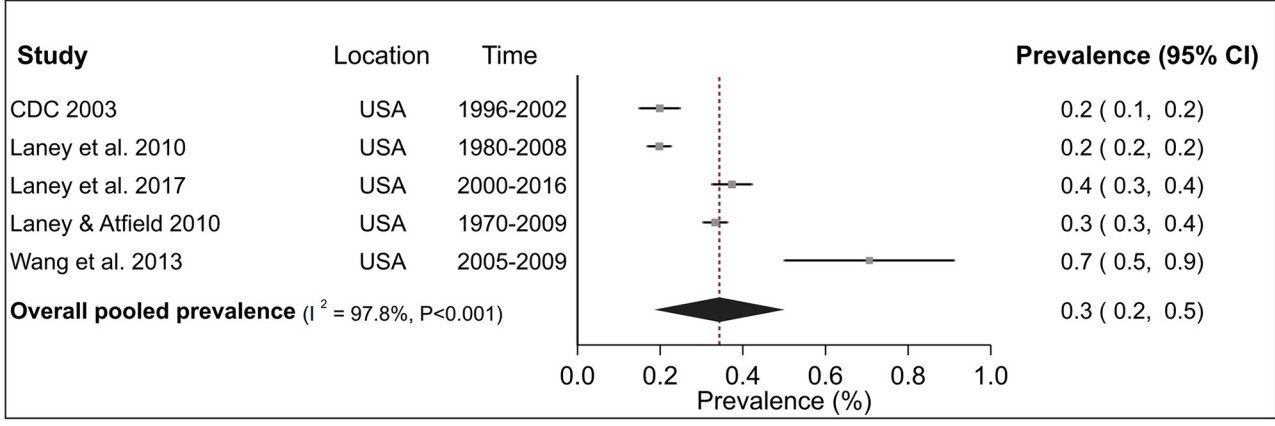

**Fig 3. Forest plot from meta-analysis for overall prevalence of progressive massive fibrosis (PMF) in the United States (USA).** The red dashed line corresponds to the overall effect size to guide the eye.

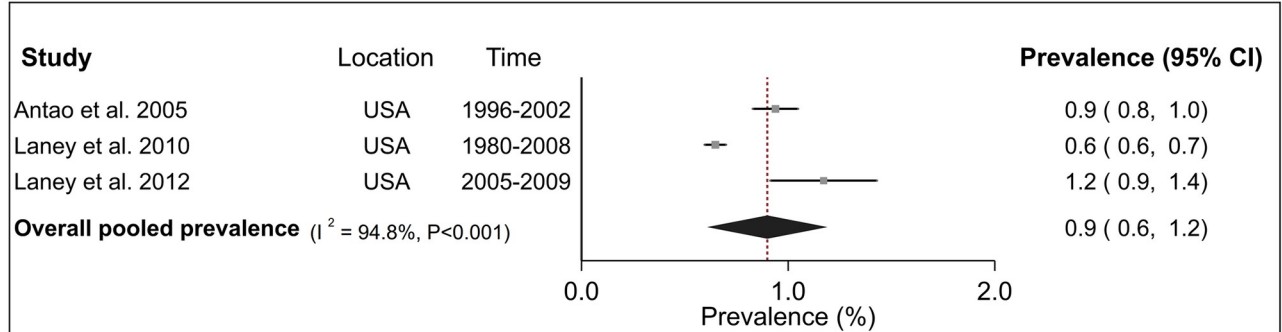

**Fig 4. Forest plot from meta-analysis for overall prevalence of advanced coal workers pneumoconiosis in the United States (USA).** The red dashed line corresponds to the overall effect size to guide the eye.

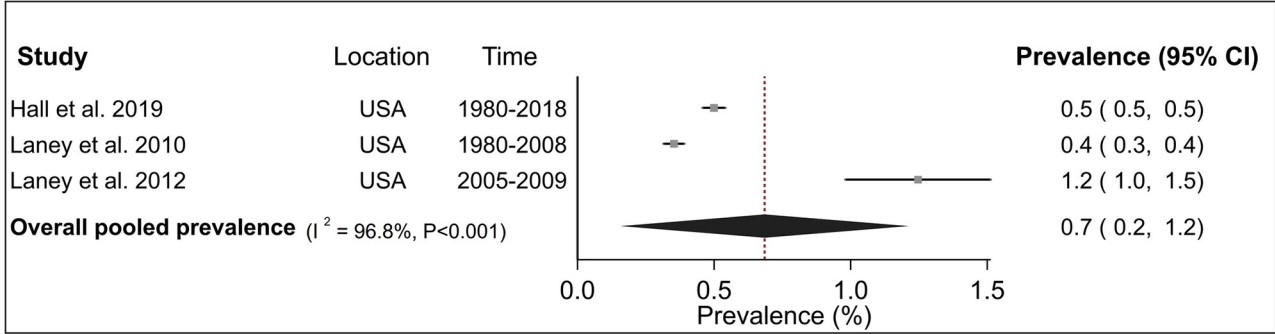

**Fig 5. Forest plot from meta-analysis for overall prevalence of r-type opacities in the United States (USA).** The red dashed line corresponds to the overall effect size to guide the eye.

There was high and significant heterogeneity between studies for All USA ($I^2$ = 98.3%, P <0.001) and regional subgroup of Central Appalachia ($I^2$ = 95.3%, P <0.001) although estimates were robust to influence of individual studies (S4 Table). No evidence of between study heterogeneity was found for 'Rest of USA' ($I^2$ = 0.0%, P = 0.978).

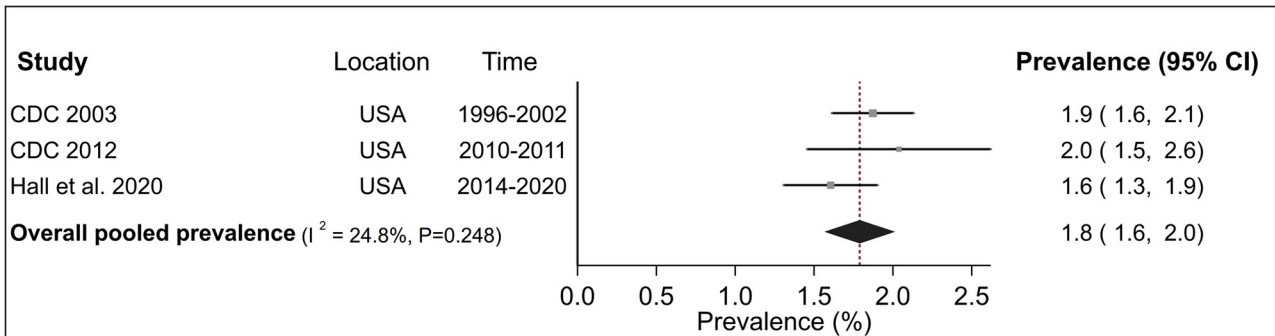

**Fig 6. Forest plot from meta-analysis for overall prevalence of coal worker's pneumoconiosis (CWP) among surface coal miners in the United States (USA).** The red dashed line corresponds to the overall effect size to guide the eye.

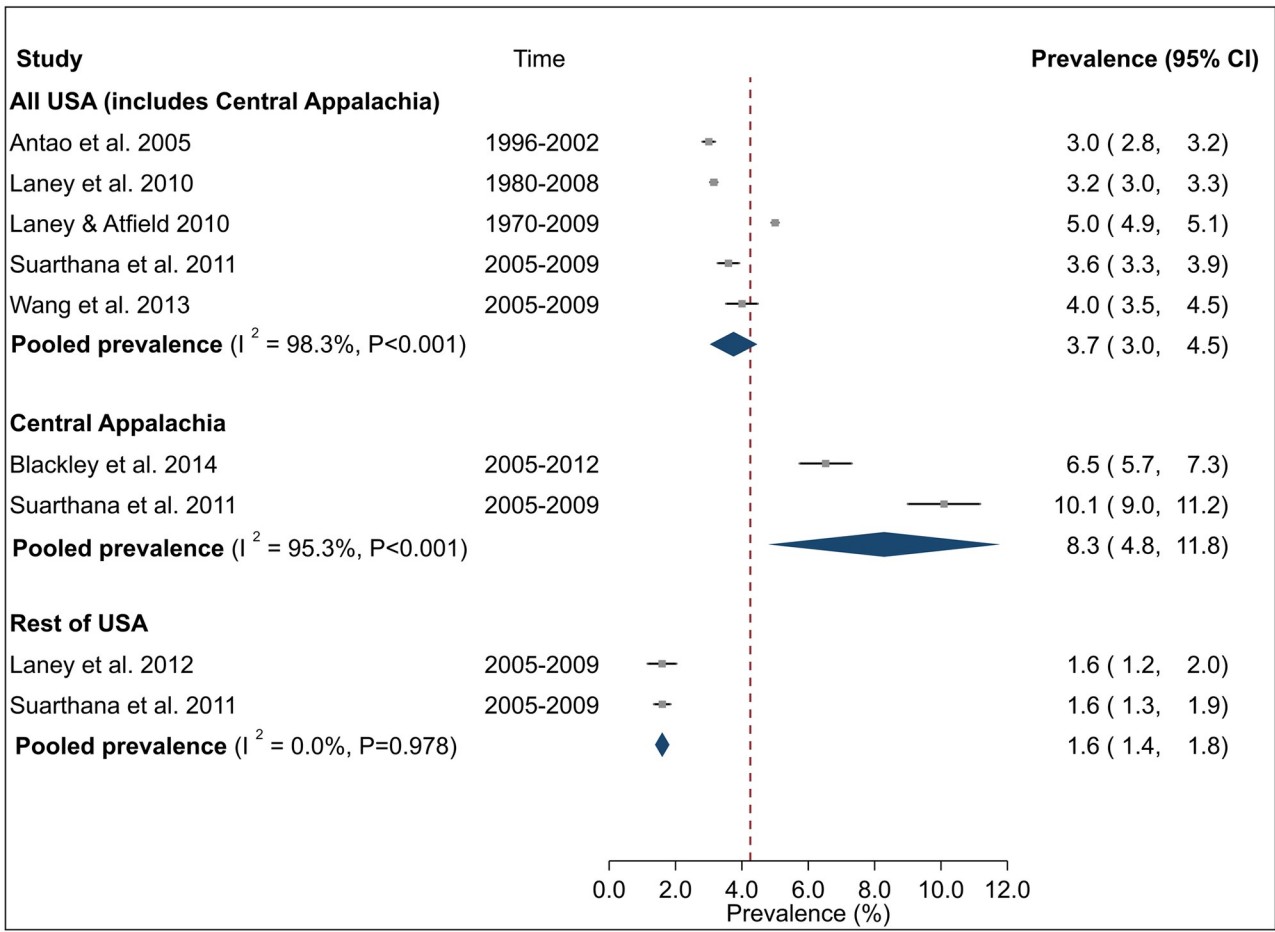

**Fig 7. Forest plot from meta-analysis for overall prevalence of coal worker's pneumoconiosis (CWP) in the United States by region: All USA, Central Appalachia and Rest of USA.** The red dashed line corresponds to the overall effect size to guide the eye. The term Central Appalachia refers to the region covered by states of Kentucky, Virginia, and West Virginia.

Sensitivity analysis by study quality (S4 Fig) gave a pooled CWP prevalence of 3.7% (95% CI 2.8–4.6%) over four high quality (low risk of bias) studies, with the point estimate being similar to the original analysis which did not exclude studies on basis of their quality.

**By study time period.** Sub-group analysis by time period indicated that the prevalence of CWP in the USA was 3.0% (95% CI 2.8–3.2%) in the 1990s over two studies and 3.6% (95% CI 3.3–4.0%) in the 2000s over three studies (S5 Table). Tests for between-study heterogeneity found no evidence of heterogeneity for the 1990s ($I^2$ = 0.0%, P = 0.901) and high, significant heterogeneity ($I^2$ = 82.5%, P = 0.004) for the 2000s. For PMF, the pooled prevalence was 0.2% (95% CI 0.1–0.2%) across two studies in the 1990s with moderate but non-significant heterogeneity ($I^2$ = 63.5%, P = 0.097) between studies (S5 Table). The corresponding estimate was 0.4% (95% CI 0.2–0.7%) across three studies during the 2000s with high and significant heterogeneity ($I^2$ = 95.4%, P <0.001).

Finally, the pooled CWP prevalence among underground miners in the USA between 2005–2009 (two studies for each region) was 3.8% (95% CI 3.4–4.1%) nationally, 8.2% (95% CI 4.6–11.8%) among miners from Central Appalachia and 1.6% (95% CI 1.4–1.8%) for the rest of the country (S5 Fig). The between-study heterogeneity was moderate but non-significant

for All USA ($I^2$ = 46.7%, P = 0.171), high and significant for Central Appalachia ($I^2$ = 96.3%, P <0.001) and non-significant for 'Rest of USA' ($I^2$ = 0.0%, P = 0.978) (S5 Fig).

No publication bias was detected for any of these meta-analyses (Egger P > = 0.50).

## Mortality

Cumulative exposure to coal mine dust was associated with increased mortality from pneumoconiosis or COPD at long-term follow-up of population-based cohorts of coal miners compared to the age-matched general male population in the USA [41, 45] and the UK [51, 52] (Table 7). Crude fatality rate for pneumoconiosis among coal miners in the USA was estimated

**Table 7. Summary of included studies for mortality or survival due to coal mine dust lung disease.**

| Author, year | Country | Period | Data source | Type of CMDLD | Number of deaths (population) | Findings |
|---|---|---|---|---|---|---|
| Attfield & Kuempel 2008. [41] | USA | 1969–1993 | Death registry[b] | Pneumoconiosis (CWP, silicosis and unspecified), | 383 (of 8,899) | SMR: (pneumoconiosis) 308 (95% CI 278–341, $p < .001$) |
| Graber et al 2014. [45] | USA | 1969–2007 | Death registry[c] | Pneumoconiosis (CWP and silicosis), COPD | Pneumoconiosis 403; COPD, 309 (of 9,033) | SMR: (pneumoconiosis) 79.7 (95% CI 72.1–87.7); (COPD) 1.11 (95% CI 0.99–1.24) |
| Beggs et al. 2015 [42] | USA | 2003–2013 | Death registry[d] | Pneumoconiosis, CWP | NS | ASR (per million)[e]: pneumoconiosis, 6.8; CWP, 1.3 ($p<0.001$) |
| CDC 2010 [44] | USA | 1968–2006 | Death registry[d] | CWP | 28,912 | Deaths/year[f]: (1968–1972) 1,106.2, (2002–2006) 300.0 ($p < .001$) ASR (per million)[g]: (1968) 1.78, (2006) 0.07; ASR[7]: (1968) 6.24, (2006) 1.02 |
| Bell & Mazurek. 2020 [43] | USA | 1998–2018 | Death registry[d] | CWP | 11,203 | Deaths: (1999) 1,002, (2018) 305.0 ($p<0.05$) ASR (per million)[h]: (1999) 4.7, (2018) 1.0 (APC 1999–2018–8.6%, $p<0.05$) |
| Coggon et al. 2010 [51] | UK | 1979–2000 | Death registry[i] | CWP, silicosis, COPD | CWP, 1,440; silicosis, 213; COPD, 10,489 | Excess deaths per year (1991–2000): CWP 49.8; Silicosis 5.0, COPD 82.6 |
| Miller & MacCalman. 2010 [52] | UK | 1959–2005 | Death registry[i] | CWP, silicosis, COPD | CWP, 222; silicosis, 10; COPD, 849 | SMR: (COPD)[j] 128.5 (95% CI 115.6–142.9) |
| Smith et al. 2006 [56] | Australia | 1979–2002 | Death registry[k] | CWP, silicosis | NS | ASR (per million males): CWP, 0.1[l] silicosis, 1.0[m] |
| Han et al. 2017 [49] | China | 1963–2014 | Hospital records | CWP | 80 (of 459) | CFR: 17.4%, |

APC, annual percentage change; ASR, age-standardised rate; CDC, Centre for Disease Control and Prevention; COPD, chronic obstructive pulmonary disease; CWP, coal workers' pneumoconiosis; CFR, case fatality rate, CI, confidence interval; NS, not stated; SMR, standardised mortality ratio; UK, United Kingdom; USA, United States

[a] Based-on cause of death codes in death certificates

[b] Social Security administration death data, Internal Revenue service records, Department of Vital Statistics

[c] National Death Index

[d] National Centre for Health Statistics' multiple cause of death files

[e] ASR in 2013

[f] aged ≥25 years

[g] aged 25–64 years

[h] aged ≥65 years

[i] Office of National Statistics

[j] 1990–2005, not reported for CWP or silicosis

[k] Australian Bureau of Statistics national mortality data set

[l] ASR per million males in 1994–1996

[m] ASR per million males in 1997–1999

to be 4.3% (1969–1993) [41] and 4.6% (1969–2007) [45]. Corresponding estimate in the UK (1959–2005) was 1.6% [52].

Three studies from the USA reported steadily declining CWP mortality rates among the general population over recent decades both nationally [42–44] and in the Central Appalachian region [43] with similar patterns also found in Australia [56]. Only one study looked at survival and found that 80 out of 459 CWP cases without tuberculosis diagnosed from 1963–2014 in a state-owned mine in China had died over the study period [49], corresponding to a mortality rate of 17.4%. The average survival time for CWP patients without pulmonary tuberculosis in this cohort was 37.9 years.

Summary estimates by mortality were not generated due to the wide variability in reported outcome measures and/or the lack of standard errors (or 95% CI) provided by the studies.

## Discussion

### Summary of main findings

To the best of our knowledge, this is the first systematic review and meta-analysis on global prevalence of CWP and other CMDLD-spectrum diseases. Combining the published prevalence estimates across individual studies through meta-analysis allowed us to increase the sample size and thereby improve the precision of the summary estimates. Thirty-one articles on CMDLD prevalence were included in this review of which 15 were retained for meta-analysis. Excluded articles typically either reported estimates for population sub-groups only, did not present sufficient information to be included or could not be combined with other studies due to wide variability in data sources.

Most studies (24 out of 31) reporting the prevalence of CWP or other CMDLD were from the USA, with a very limited number of studies being available from other countries. This also meant that 12 of 15 studies included in the meta-analysis were from the USA limiting our ability to understand the contemporary burden of CMDLD in other coal-producing countries.

Combining the study-specific estimates through meta-analysis indicated that the pooled prevalence of CWP among underground miners was 3.7%, although confidence intervals were wide. Corresponding estimates for PMF, advanced CWP and r-type opacities in the USA were 0.3%, 0.9% and 0.7% respectively. These estimates highlighted the substantial burden of pneumoconiosis among coal miners and are suggestive of inadequate dust control standards and a possible need for more stringent limits.

Given the high heterogeneity between studies, the pooled prevalence estimates should be interpreted with some caution because they may not be representative of specific countries, particularly if the standards for dust control vary between countries [57]. However all included studies used the standardised ILO diagnostic criteria [9] for defining cases of CWP, PMF, silicosis and r-type opacities in similar populations (adult current/and or former coal miners) thereby ensuring some comparability. Moreover, prevalence estimates from the USA were only combined if they were drawn from the same type of population. In practise this meant that studies based on compensation databases for example were excluded as denominator was not comparable to population-based surveillance databases of coal miners. We found no evidence of publication bias or bias related to study quality (as measured by risk of bias).

Studies using USA national surveillance data may have underestimated the true population-based prevalence of pneumoconiosis among all coal miners. It is unlikely that the reported surveillance data capture all disease cases among coal miners given that these programs are voluntary, restricted to active coal miners, and may not incorporate the long latency

between dust exposure and CWP development [33]. However, studies using national surveillance data in the USA found no link between participation or other selection factors and the changes in CWP prevalence [33]. In contrast, studies using federal black lung benefits program claims data [7, 15] would have overestimated CWP prevalence because instead of all active coal miners, their denominator only included active or former coal mine workers who had applied for benefits for a suspected CMDLD. Regardless of the data source, all included studies in this review consistently reported that CWP prevalence among miners in the USA had increased in recent decades.

## Changes over time in CWP/PMF prevalence

Several of the included studies in this review stated that following decades of declining prevalence in the USA, there was an increase in pneumoconiosis prevalence and severity among underground coal miners nationwide since the late 2000's, even among miners who worked exclusively under current dust exposure limits [7, 15, 22, 32, 36]. In particular, this pattern was reported for miners from the Central Appalachian coal mining region [22, 35, 38] and those with at least 20-years tenure [22, 35, 38].

Sub-group analysis by time period suggested that pooled prevalence of CWP in the USA was higher in 2000s than 1990s, whereas for PMF although the summary point estimate was suggestive of a similar pattern, confidence intervals were overlapping. Moreover, the pooled prevalence of CWP was around 80% higher in the Central Appalachia region than the rest of the USA from 2005–2009. The limited number of studies however made it difficult to draw definitive conclusions on the magnitude and extent of any temporal changes in CWP (of any severity) or PMF prevalence.

Proposed contributors included increased dust exposure reflecting changes in working practice and mining techniques, changes in dust composition, poorer compliance to current exposure standards and inadequate regulatory levels [1, 2, 6, 15, 21, 38, 58]. Advances in dust exposure control technology and equipment may not be keeping pace with contemporary mining techniques that can potentially expose workers to higher concentrations of dust [2, 58]. Identifying and developing engineering dust controls and real-time dust monitoring capabilities that can then be implemented in the coal mining industry is crucial to reducing respirable dust exposure.

Given that all included studies defined CWP cases based on the ILO diagnostic criteria [9], reported prevalence estimates were unlikely to have been impacted by more sensitive diagnostic techniques such as high-resolution computed tomography screening [1].

The reported changes in prevalence of CWP and/or PMF in the USA may also reflect changes in the age range of coal miners over time and/or between geographical locations with older age increasing risk of disease. Our ability to consider the impact of age on reported prevalence estimates was limited by the lack of prevalence estimates stratified by age group across the included studies.

The limited number of studies meant that it was unclear whether the prevalence of pneumoconiosis had also changed over time in in countries other than the USA. However, based on these studies, prevalence of CWP reported from China [46–48], South Africa [53] and Turkey [54] was not higher than those reported from USA studies based on national surveillance data [19, 25, 31, 32, 34, 35, 37–39], while lower prevalence was reported in the UK [50]. Moreover, sub-group analysis by country indicated that the pooled point prevalence estimates of CWP in USA and China were very similar.

Any reduction of dust suppression activities and decreased vigilance regarding cumulative inhalation dust exposure could potentially lead to significant increases in CMDLD diagnoses

and morbidity. In addition, legislated standards for dust levels vary globally and the evidence regarding effectiveness of existing regulatory controls in preventing CMDLD is currently not available [1, 2, 6], making the interpretation of international comparisons difficult.

## Mine characteristics and CWP prevalence

In the USA, CWP prevalence was consistently higher among miners with longer tenure [25, 38] and those working in mines with a smaller workforce [21, 25] possibly as a result of increased and longer duration of dust exposure, both of which are associated with increased risk of CWP [3, 58]. Moreover, there may be differences in work practices and conditions by mine size as smaller mines may lack resources to ensure adequate dust reduction [21, 32, 35]. Further research is required to better understand how these factors are associated with CWP prevalence.

Although the summary estimate of CWP prevalence among surface miners was around 51% lower than for underground miners in the USA (1.8 versus 3.7%), it was not negligible. Pneumoconiosis cases among working surface miners, especially those with no underground mining experience, has been attributed to excessive exposure to respirable silicosis dust during drilling or blasting operations [28, 30].

## Regional variation in CMDLD prevalence (USA only)

There was a consistently higher burden of CWP [22, 28, 35, 38] and/or its most severe form (PMF) [21, 26, 35] among contemporary miners in the Central Appalachian region compared to other coal mining regions in the USA. These regional patterns may be reflected in the summary pooled estimates for CWP among underground miners in the USA which were significantly higher for Central Appalachia than the 'Rest of USA' or nationally. While definitive reasons for these regional patterns are unknown, commonly proposed contributors from included studies include regional differences in mining practice, type of coal excavated, safety culture and geology [6, 28, 38]. It is also possible that current dust exposure limits may need to be reduced and/or there is inadequate compliance with current guidelines [6, 22, 28, 35, 38].

## Prevalence estimates for other CMDLD

The reported prevalence of COPD among underground miners in the Ukraine of 14.9% [55], was about double that of the prevalence found in the adult population over the same time period but was similar to that reported for smokers [59]. Given that smoking is the primary risk factor for COPD, these reported estimates are likely to have been impacted by confounding through smoking, especially given around 62% of the Ukraine study sample were current smokers.

## Australian context

Our interest in carrying out this review was primarily driven by recent reports of CWP cases in Australia [60, 61], another major coal producing country [62]. Nonetheless, we found no published prevalence estimates for CWP or any other CMDLD in Australia. In contrast to the United States, for example, there is no national comprehensive reporting system in Australia [3]. However, on 1 July 2019 mandatory reporting of dust-related lung diseases came into effect in the state of Queensland (Australia) [63]. The lack of a national perspective on the current CWP burden in Australia is of concern, especially given the international literature, as summarized in this review, provides evidence supporting an increasing prevalence of this disease.

There are mandatory surveillance programs for coal miners in some Australian states such as the Coal Mine Workers' Health Scheme in Queensland [64], and the 'Order 43 Coal Mine Workers Medicals' in New South Wales (since July 2018) [65] which require an initial assessment on entering the industry followed by mandatory periodic assessments while an active coal miner.

Improved data collection and management of periodic surveillance data for coal workers, enhanced surveillance methods (such as computed tomography) and establishment of a comprehensive occupational lung disease registry nationwide have been proposed as measures to provide evidence-based understanding of current (and future) CMDLD burden in Australia [3, 66]. In addition, dust regulatory controls in Australia were less stringent than USA standards [3] until September 2020, when they were revised in Queensland and New South Wales to the same limits as those in the USA [5].

## Mortality/Survival

There was limited information on global mortality for CMDLD. Reductions in CWP mortality rates were reported in the USA and Australia [42–44, 56], however these are in terms of mortality among the whole population, rather than specifically coal miners. This decreasing trend in CMDLD mortality at a population level was also observed in the Central Appalachian states (USA) despite the increase in CWP prevalence over recent years [42]. The main reason for this discrepancy may be the time between diagnosis and death, in that recently diagnosed cases are not yet accounted for in the latest mortality data. Trends in CMDLD mortality rates based on the total population will not account for any changes in the proportion of that population who have worked as coal miners.

These population mortality trends are also inconsistent with other studies that have reported the potential years of life lost among coal miners who died of CMDLD (hence not included in this review). Two USA studies [44, 67] found that the mean potential years of life lost had increased over time, which is consistent with either an increase in deaths or deaths at an earlier age. While these measures are useful, to enable valid comparisons of mortality rates between population groups, reporting consistent measures of mortality rates is crucial to better understand inequalities and changes in the CMDLD mortality burden. This includes measuring mortality rates in terms of the total population, but also mortality specifically among the cohort of coal miners.

## Limitations

Limitations included high heterogeneity between the retained studies for the prevalence meta-analyses and the under-representation of studies from countries other than the USA. Of 15 studies retained for calculation of pooled estimates, 12 were from the USA and three from China. No studies were found from major coal producing countries in Asia other than China or other high-income countries producing coal such as Australia and Germany [62]. The heterogeneity in pooled prevalence estimates may reflect the variability in study time periods, data collection and inherent bias of the data sources [33]. However, there are also other factors such as differences in geology, working practices, safety culture and operational issues impacting enforcement of dust standards that may have contributed to inter-study heterogeneity but are not necessarily limitations. Sampling and other methodological biases in primary studies also cannot be excluded.

A lack of studies from countries besides the USA limited our ability to make comparisons between countries. The small number of studies also restricted exploration of the impact of factors such as study time period, regional location, mine characteristics including size or type

and mining tenure on the pooled prevalence estimates in the USA itself. In particular, our ability to effectively summarize evidence for changes in prevalence of CWP over time was restricted by the wide heterogeneity in study time periods. This only allowed a small number of studies to be combined in sub-group analysis for any specific time period.

Despite the increasing prevalence of CMDLD in the modern era, studies assessing the impact of a CMDLD diagnosis on coal miner's lives were sparse. Moreover, summary estimates of CMDLD mortality and/or survival could not be generated due to wide heterogeneity in reported outcome measures, cohort characteristics and data sources. In addition, none of the included studies provided information on standard errors (or 95% CI) for the reported estimates.

Multiple databases were searched with complex queries. Additionally, the reference lists of the identified articles and reviews were also used to identify potentially relevant articles. Nevertheless, the included articles were limited to those indexed in the databases that were searched for this review. Hence, it is possible that the search terms and criteria used, as well as selection of citation databases, could have unintentionally led to the exclusion of relevant articles. As only peer-reviewed published studies in English were included; relevant articles published in other languages may have been missed. Moreover, given the lack of defined standards for rigorously searching grey literature, limited formal archiving and standard indexing for such studies, potentially incomplete or inaccurate information on factors such as authorship, date of publication and data sources and wide heterogeneity in study completeness, quality, and consistency across studies, we purposely did not include grey literature in the current review.

## Conclusions

This study summarizes published evidence from global studies on prevalence, mortality, and survival due to CMDLD, especially CWP. Overall summary pooled prevalence estimates highlighted that CWP remains a public health concern and were suggestive of an increase in prevalence among more recent time periods. CWP prevalence was also higher among underground coal miners than surface coal miners and varied by region within the USA. The international generalisability of findings was limited by most studies being sourced from the USA, while it was difficult to draw definitive conclusions from the small number of mortality and survival studies. Comparisons of findings across studies was difficult due to the inconsistent data collection and reporting methods used.

From a public health perspective, the continuing prevalence of CWP prevalence in the modern era highlight the need for continuing surveillance and prevention efforts involving both dust monitoring and suppression in coal miners. Increased scrutiny of current dust monitoring practices, efforts to achieve global standardization of current guidelines, and ensuring compliance with effective dust exposure protections are crucial to protect miners from developing this entirely preventable occupational disease.

As CMDLD is a progressive disease with a long latency, respiratory morbidity can occur long after the end of exposure. Establishing an effective process to monitor the effectiveness of interventions to reduce the impact of respirable dust exposure on coal miners will require ongoing health surveillance over many years, including after miners have left the workforce, combined with standardised data collection and reporting systems.

## Supporting information

**S1 Checklist. PRISMA 2009 checklist.**
(DOC)

**S1 Table. Search strategies.**
(PDF)

**S2 Table. Critical appraisal tools.**
(PDF)

**S3 Table. Results of critical appraisal for studies included in systematic review.**
(PDF)

**S4 Table. Sensitivity analyses for the influence of individual studies on pooled prevalence estimates for coal mine lung disease in the United States.** Included meta-analyses are for coal workers pneumoconiosis (CWP) by disease severity (progressive massive fibrosis, advanced coal workers pneumoconiosis), mine type (surface), region (Central Appalachia versus overall United States) and r-type opacities.
(PDF)

**S5 Table. Forest plot from meta-analysis for prevalence of coal workers pneumoconiosis (top) and progressive massive fibrosis (bottom) in the United States by study time period (1990s versus 2000s).**
(PDF)

**S1 Fig. Plot of global prevalence estimates of coal mine dust lung disease by country.** Abbreviations are USA United States, UK United Kingdom, CMDLD coal mine dust lung disease, CWP coal workers pneumoconiosis, PMF progressive massive fibrosis, Advanced CWP advanced coal workers pneumoconiosis. Other CMDLD includes silicosis, lung function abnormality and chronic obstructive pulmonary disease. Other countries are UK, Turkey, South Africa and Ukraine. Within each country, studies are shown by mid-year of the study period.
(TIF)

**S2 Fig. Change in prevalence of coal mine dust lung disease among coal miners based on studies included in the systematic review.** Data was sourced from Laney & Atfield 2010 [32]; Laney & Atfield 2014 [33] Blackley et al 2018a [22], Almberg et al 2018 [15] and Scarsbrick et al 2002 [50] in the systematic review. Please note that the time periods reported is limited to those reported by included studies. Trends for CWP prevalence are shown in green and for PMF prevalence in red. Abbreviations are USA United States, UK United Kingdom, CWP coal workers pneumoconiosis, PMF progressive massive fibrosis. Central Appalachia includes the states of Kentucky, Virginia, and West Virginia in USA.
(TIF)

**S3 Fig. Sensitivity analysis for the influence of individual studies on the pooled prevalence estimates from meta-analysis for overall prevalence of coal workers pneumoconiosis among underground coal miners.**
(TIF)

**S4 Fig. Sensitivity analysis for the influence of study quality (risk of bias) on the pooled prevalence estimates for coal workers pneumoconiosis among underground coal miners in the United States.**
(TIF)

**S5 Fig. Forest plot from meta-analysis for prevalence of coal worker's pneumoconiosis (CWP) in the United States (USA) by region: All USA, Central Appalachia, and Rest of USA from 2005 to 2009.** The term Central Appalachia refers to the region covered by states of

Kentucky, Virginia, and West Virginia.
(TIF)

## Author Contributions

**Conceptualization:** Paramita Dasgupta, Jessica Cameron, Peter Baade.

**Data curation:** Cynthia Lu, Paramita Dasgupta.

**Formal analysis:** Cynthia Lu, Paramita Dasgupta.

**Funding acquisition:** Peter Baade.

**Investigation:** Cynthia Lu, Paramita Dasgupta.

**Methodology:** Cynthia Lu, Paramita Dasgupta, Peter Baade.

**Project administration:** Peter Baade.

**Resources:** Peter Baade.

**Supervision:** Peter Baade.

**Validation:** Paramita Dasgupta, Peter Baade.

**Visualization:** Cynthia Lu, Paramita Dasgupta.

**Writing – original draft:** Cynthia Lu, Paramita Dasgupta.

**Writing – review & editing:** Paramita Dasgupta, Jessica Cameron, Lin Fritschi, Peter Baade.

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
