## [Decision Letter · Decision Letter 0]

11 May 2021

PONE-D-21-09488

Global patterns in prevalence, mortality and survival due to coal mine dust lung disease: A systematic review and meta-analysis

PLOS ONE

Dear Dr. Baade.

Thank you for submitting your manuscript to PLOS ONE. After careful consideration, we feel that it has merit but does not fully meet PLOS ONE’s publication criteria as it currently stands. Therefore, we invite you to submit a revised version of the manuscript that addresses the points raised during the review process.

I have some major concerns as follows:

Introduction:

In the last paragraph, through your methodology, you cannot  solve this problem. 

Method:

Line 93: PRISMA suggested protocol registration. 

Line 202 -204 That statement may be your result, not proper in the method.

Figure 1: please remove unnecessary symbols. What does " the other sources " mean?

Table 2: please rearrange the table to make it easy to follow (The same is true for other tables). For example, all studies conducted in the USA, arrange them according the study period......

Footnote b and c are redundant as they have mentioned these in the method section.

Result:

Could you provide graphic illustration of the patten of the global prevalence and highlight most studies were conducted in the specific continents, such as in the USA. 

Please give the corresponding  citation for the studies you  mentioned. 

Line 373-381

I suggest  presenting the results from each study in a line chart or curve chart to increase visualization. 

Besides, please interpret this finding cautiously in the discussion section.

Please check the forest plot if the study included in the meta-analysis shared the overlapping study period from the same data sources. That is not justice. 

Conclusion section:

Please concisely and briefly highlight in 3-5 points of your systematic review and meta-analysis. 

We look forward to receiving your revised manuscript.

Kind regards,

Yuan-Pin Hsu

Academic Editor

PLOS ONE

Journal Requirements:

2) Please provide a table reporting in detail the results of your quality assessment, showing how each included study scored on every item of the Joanna Briggs quality assessment tool used.

Reviewers' comments:

Reviewer's Responses to Questions

**Comments to the Author**

1. Is the manuscript technically sound, and do the data support the conclusions?

Reviewer #1: Partly

Reviewer #2: Yes

Reviewer #3: Partly

Reviewer #4: Yes

2. Has the statistical analysis been performed appropriately and rigorously? 

Reviewer #1: I Don't Know

Reviewer #2: Yes

Reviewer #3: Yes

Reviewer #4: Yes

3. Have the authors made all data underlying the findings in their manuscript fully available?

Reviewer #1: Yes

Reviewer #2: Yes

Reviewer #3: Yes

Reviewer #4: Yes

4. Is the manuscript presented in an intelligible fashion and written in standard English?

Reviewer #1: Yes

Reviewer #2: Yes

Reviewer #3: Yes

Reviewer #4: Yes

5. Review Comments to the Author

Reviewer #1: The authors present "global" data on CWP based on metanalysis and pooling of data from multiple studies over many years ( with most of the studies being US ones)

Major Concerns:

The temporal changes described in the paper and critically important to understanding what is happening with CWP in the US are not apparent in any of the figures- which present data for each study that may cover over 40 years- during which the prevalence and outcomes of CWP may have changed dramatically. I think the authors need to determine a way to break the data into 5 or 10 year intervals for presentation in the figures so that equals can be better compared to equals. Many of he primary papers cited have presented data in 5 or 10 year intervals- so this should be possible.

Reviewer #2: This systematic review and meta-analysis aimed to evaluate the prevalence and the mortality due to Coal mine dust lung disease world wide. The authors achieved to give global prevalence data based on the available literature but were unable get an overall figure for mortality or survival due to the heterogeneity and non standardized way of estimation in the few studies that have addressed the question.

As often in meta-analysis and in particular in this case, the conclusions are limited by the quality and representativeness of the original publications. The authors have made their best to extract meaningful and comparable results from the available studies, but were faced, among other hurdles, with a geographical bias with overrepresentation of studies from the USA.

The main message of this meta-analysis is a prevalence of CWP of 3% with a wide range from 0.2 to 5.5%. Another important message highlited in the discussion is the increase of CWP prevalence over time, in particular from the 2000’s in the USA. This affirmation was mainly made on the quoted studies, however the data presented on Table 3 does not show striking higher prevalence for more recent period. For example, ref 30 that covers the period 2011-2014 has a low 0.2% prevalence whereas ref 25 shows a 3% prevalence for an earlier period. If the authors want to keep this as a major message of the present review, they should find a way to present this recent trend in a more convincing way. Discussion should also be made on the possibility for better or earlier diagnosis in recent studies due to more systematic screening such as the use of CT scans.

I do not understand the forest plot of figure 7: Do the studies listed under “All USA” include or exclude data from central Appalachia below?

Table 7 on mortality and survival is particularly frustrating to read. At least 5 different ways of evaluating mortality have been used making any comparison between studies a nightmare. Moreover, the type of CMDLD accounted for the cause of death varied between studies, some of them including COPD. I understand that due to this wide methodologic variability summary estimates could not be generated. However, I found a bit limited to conclude only with the statement that CWP is associated with an increased in mortality, which is common knowledge. I wonder if a crude case-fatality rate comparison would be possible and informative ?

Ref 17 is the only study assessing the prevalence of COPD in the miners. The data of 14.6% is about the double of the prevalence found in the adult population, but is similar to the prevalence reported for active smokers. Besides the exact definition of COPD applied to such survey, the confounding factor of smoking should of course be accounted for. Although COPD was not the main topic of this review a short discussion on these limitations would be useful.

Reviewer #3: Coal workers’ pneumoconiosis (CWP), also known as black lung disease, is caused by long-term exposure to coal dust. Coal miners are vulnerable to CWP, as their work environment is highly exposed to coal dust. Many coal miners are likely to develop CWP due to an increase in coal production and utilization, especially in developing countries. CWP with other diseases including mixed dust pneumoconiosis, silicosis comprises the Coal mine dust lung disease (CMDLD). The authors of this manuscript went a great length in collecting and controlling the quality of data, which to reduce biases and to remove duplication, to perform a systematic review and meta-analysis of published research on prevalence, mortality, and survival for CMDLD. Although the objective might be a little broad, the objective is still clear. The pipeline of the authors research is well thought of and the meta-analysis is reasonable in which in the end forest plots were used to show each studies effect and significance was also clear.

As the authors have mentioned biases still may exist in the results, however, is understandable as such biases are not limited to the authors research but a common concern in research conducting meta-analysis. Although such biases might still be a concern, the authors have well stated it in the limitation section of their manuscript and took account for it in interpreting their results. The authors interpretation of their results was interesting, and the logic was reasonable although there would need more follow up research to conclude in their findings. For that reason, the authors well positioned their interpretation in the discussion section in which the authors research may promote and contribute to future research, which is much needed in directing government policies for the health and welfare of coal miners.

Suggestion to the study (please note the authors have no need to address or follow up on the suggestions): The manuscript would have been more enriched if the authors would have gone another depth in considering factors such as the age range of the minors of each study, as the age range of the minors might have changed during periods of time and different between locations in which older age might increase the risk of the disease. Although the authors consider other diseases of CMDLD, the authors mainly focus on CWP, which is reasonable, however, for the objective of the study other meta-analysis on related diseases such as Anthracosis would have provided a greater view. The authors do not have to address these suggestions as the limitation of data availability is understandable and the suggestions may be subjects for other studies and is not the focus of this manuscript.

One minor concern that the authors should consider and address: Due to the limitation of the data focusing on the studies conducted in the United States was not only reasonable but strategical and effective in addressing the objective of this study that may contribute to the national perspective on the current CWP burden in Australia and furthermore, guiding the Australian government policies for the health and welfare of coal miners. However, one minor concern would be as the studies used for the meta-analysis are mainly from the United States (n = 29) and the second is only followed by 4 studies from China, using the term ‘Global patterns’ in the title might be misleading and is a slight concern. To the best of my knowledge the patterns that are emphasized in the manuscript and the logical deduction the authors made are focused on studies conducted in the United States. Except for the fact that studies from 7 countries were used for the meta-analysis if there is no statistical evidence the pattern the authors emphasize is global, I would suggest the authors to consider rephrasing, which could be done by doing just one of the following.

1) Clarify or summarize what were the global patterns observed and then emphasize the patterns observed in the United states, as the patterns referring in the title and patterns emphasized in the discussion might be confused.

2) Consider rephrasing the title, and if the term ‘global’ is still preferred, it would be recommended to rephrase the title for example something like ‘A systematic review and meta-analysis on global studies of ~ ’.

Reviewer #4: This is an important and well written manuscript reporting results from a well conducted meta-analysis of prevalence estimates for the diseases which encompass coal mine workers lung disease (CMWLD). It is the first systematic review and meta-analysis on the global prevalence of coal workers pneumoconiosis (CWP) and other CMWLD-spectrum disorders. These findings have important public health implications as they highlight the substantial burden of CMWLD and are suggestive that dust control standards have failed miners historically and need more stringent limits and enforcement.

The strengths of this study include: that it followed the PRISMA guidelines; the thorough search for articles; the review and data abstraction processes which included robust quality control procedures; and, the assessment of potential bias using the Joanna Briggs Institute (JBI) critical appraisal tools. The main issue, which ais addressable, is that prevalence estimates were combined despite significant study heterogeneity. The authors justify this by stating that all studies included in the meta-analysis used the standardized International Labour Organization (ILO) diagnostic criteria. While this is very important, an equally important criterium for combining prevalence estimates is that they be drawn from the same type of population, for example all from the general population or all from a population of coal miners.

Major comments:

• Combining estimates in the face of significant heterogeneity is ill advised. Three instances of this are:

1. There appears to be significant heterogeneity among the studies included in the analysis of overall prevalence of coal worker’s pneumoconiosis (CWP) among underground coal miners. (Figure 2) - the source may be combining studies from different regions of the world so the degree to which this meta-prevalence is generalizable to a ‘global population’ is unclear.

2. The study by Arif AA et. al. (2020) estimated population rates (county-level) of CWP while the other studies in the group it is combined with used surveillance methodologies so their denominator is coal miners. These should not nbe in the same meta-analysis.

3. The authors note that the prevalence estimates from the studies using workers compensation program data and those using surveillance data likely are from very different populations. T(he population included the workers compensation program are likely skewed toward those with longer mining tenure and disease symptoms, and so likely overestimate the prevalence of CMWLD.) In the forest plots in Fig 3 the using workers compensation program data (Graber et al, 2017 and Almberg et al., 2018, respectively) are clear outliers. As such they should not be combined with the other studies in the group that are more population -based surveillance methodologies. This is further supported by the sensitivity analyses which “indicated that pooled estimates were particularly sensitive to the study by Graber”)

• The Graber et al 2014 study (1969/71- 2007) is an update of Attfield & Kuempel 2008 (1969/71- 1993) and the same outcomes were included in the metanalysis (pneumoconiosis (CWP, silicosis, COPD) so the reason for inclusion of both is unclear.

• Given that the prevalence of CMWLD has changed significantly over the period of studies covered (1959 and 2019), the authors should consider stratifying the analyses by time if sample size permits

• The authors do an impressive bias assessment using the JBI tool and present robust results. Curiously, they did use this information to inform the ananysis. Sensitivity analyses based on the findings would be informative.

Minor comments:

• Beginning line 70 “The increase in CWP prevalence has occurred even among miners who were subject to regulatory guidelines their entire working life. Consider whether citation #15 is also appropriate here

• Beginning line 151: please clarify that these are states with the United States

• Line 116 (and Line 28) , Web of Science is included as a database but is not included in Table S1

• Beginning line 194: Please clarify how homogeneity of study characteristics was assessed

6. PLOS authors have the option to publish the peer review history of their article (what does this mean?). If published, this will include your full peer review and any attached files.

Reviewer #1: **Yes: **David Mannino

Reviewer #2: No

Reviewer #3: **Yes: **Seung Han Baek

Reviewer #4: No

---

## [Author Response · Author response to Decision Letter 0]

30 Jun 2021

PONE-D-21-09488

Global patterns in prevalence, mortality and survival due to coal mine dust lung disease: A systematic review and meta-analysis

• All references given in this document are in the Author, Year format to avoid confusion with the references in the manuscript. 

• Please note that all Page Numbers refer to the revised manuscript without track changes. 

Responses to comments

Editor

1. Introduction: In the last paragraph, through your methodology, you cannot solve this problem. 

Authors’ response: We have revised this text for clarity: (Introduction, Page 4, Lines 80-84): 

This limits the ability to quantify the historical trends and current disease burden and to direct policy and prevention to the most appropriate areas. In an attempt to quantify historical and current burden of disease, in this study, we systematically reviewed published international estimates of prevalence, mortality, and survival for CMDLD and derived summary measures through meta-analyses.

2. Method: Line 93: PRISMA suggested protocol registration. 

Authors’ response: As a protocol can only be registered before commencing a systematic review, this protocol cannot be registered retrospectively. The following has been added to Methods (Protocol, Page 5, Lines 94-96,):

A protocol for this systematic review and meta-analysis was not registered or published prior to this work commencing and has not been subsequently submitted since protocols cannot be registered retrospectively.

3. Line 202 -204 That statement may be your result, not proper in the method.

Authors’ response: These sentences are describing why a meta- analysis could not be carried out for mortality/survival papers, so we felt that these sentences need to be in Methods for clarity.

No changes have been made to the manuscript.

4. Figure 1: please remove unnecessary symbols. What does " the other sources " mean?

Authors’ response: The symbols have been removed.

The term ‘other sources’ refers to references obtained from the reference list of review articles.

The following has been added (Results, Study selection, Page 11, Lines 236-238):

As shown in a PRISMA diagram (Fig 1), a total of 405 articles were identified through the search queries with two more identified through other sources (that is, references identified through the reference lists of review articles). 

Fig 1 PRISMA flowchart for study selection 

5. Table 2: please rearrange the table to make it easy to follow (The same is true for other tables). For example, all studies conducted in the USA, arrange them according the study period.

Authors’ response: Given the substantial overlap in time periods between studies, ordering the papers by study period would not provide much greater clarity. However, we have reordered the tables by author within country for each type of disease or assessed outcome.

Additional commentary has been added to the Methods (Data extraction, Page 7, Lines 156-157):

For each type of disease (Table 1) studies were sorted alphabetically by author within country for each outcome measure.

6. Footnote b and c are redundant as they have mentioned these in the method section.

Authors’ response: Although mentioned in the text, the material included in the Tables need to be self-explanatory. Hence, we prefer to keep them for clarity.

No changes have been made to the manuscript

7. Result: Could you provide graphic illustration of the patten of the global prevalence and highlight most studies were conducted in the specific continents, such as in the USA. 

The following Figure has been added to Supplemental Information

S1 Fig Plot of global prevalence estimates of coal mine dust lung disease by country. Abbreviations are USA United States, UK United Kingdom, CMDLD coal mine dust lung disease, CWP coal workers pneumoconiosis, PMF progressive massive fibrosis, Advanced CWP advanced coal workers pneumoconiosis. Other CMDLD includes silicosis, lung function abnormality and chronic obstructive pulmonary disease. Other countries are UK, Turkey, South Africa and Ukraine. Within each country, studies are shown by mid-year of the study period. 

The following commentary has been added to the manuscript (Results, Study characteristics, Page 12, Line 256-258): 

Patterns of prevalence for CMDLD by key disease types and country across the 31 included studies in this systematic review are shown graphically in S1 Fig.

8. Please give the corresponding citation for the studies you mentioned

Authors’ response: Corresponding citations have been added for 40 included studies (Results, Study characteristics, Page 12, Lines 250-252).

9. Line 373-381 suggest presenting the results from each study in a line chart or curve chart to increase visualization. 

Authors’ response: An additional Figure has been included in Supplemental Information (S2 Fig):

S2 Fig Change in prevalence of coal mine dust lung disease by time based on studies included in the systematic review. 

Data was sourced from Laney & Atfield 2010 (Laney and Attfield 2010); Laney & Atfield 2014 (Laney and Attfield 2014) Blackley et al 2018a (Blackley, Halldin et al. 2018), Almberg et al 2018 (Almberg, Halldin et al. 2018) and Scarsbrick et al 2002 (Scarisbrick and Quinlan 2002). Please note that the time periods reported are limited to those reported by included studies. Trends for CWP prevalence are shown in green and for PMF prevalence in red. Abbreviations are USA United States, UK United Kingdom, CWP coal workers pneumoconiosis, PMF progressive massive fibrosis. Central Appalachia includes the states of Kentucky, Virginia, and West Virginia in USA.

The following has been added to manuscript (Results, Changes over time in CWP prevalence, Page 20, Line 401: 

The prevalence of CWP was also found to vary over time (S2 Fig).

10. Besides, please interpret this finding cautiously in the discussion section.

Authors’ response: Changes have been made to the Discussion that also reflect responses to other comments made by the reviewers (Reviewer 1 #14, Reviewer 2 #16, and Reviewer 4 #32). 

Details of these changes are given below (Discussion, Changes over time in CWP prevalence):

(Page 31, Lines 636-641) Several of the included studies in review stated that following decades of declining prevalence in the USA, there was an increase in pneumoconiosis prevalence and severity among underground coal miners nationwide since the late 2000’s, even among miners who worked exclusively under current dust exposure limits (Laney and Attfield 2010, Graber, Harris et al. 2017, Laney, Blackley et al. 2017, Almberg, Halldin et al. 2018, Blackley, Halldin et al. 2018). In particular, this pattern was reported for miners from the Central Appalachian coal mining region (Suarthana, Laney et al. 2011, Laney, Petsonk et al. 2012, Blackley, Halldin et al. 2018) and those with at least 20-years tenure (Suarthana, Laney et al. 2011, Laney, Petsonk et al. 2012, Blackley, Halldin et al. 2018). 

(Page 32, Lines 642-648) Sub-group analysis by time period suggested that pooled prevalence of CWP in the USA was higher in 2000s than 1990s, whereas for PMF although the summary point estimate was suggestive of a similar pattern, confidence intervals were overlapping. Moreover, the pooled prevalence of CWP was around 80% higher in the Central Appalachia region than the rest of the USA from 2005-2009. The limited number of studies however made it difficult to draw definitive conclusions on the magnitude and extent of any temporal changes in CWP (of any severity) or PMF prevalence. 

(Page 32, Lines 649-651) Proposed contributors included increased dust exposure reflecting changes in working practice and mining techniques, changes in dust composition, poorer compliance to current exposure standards and inadequate regulatory levels (Suarthana, Laney et al. 2011, Blackley, Halldin et al. 2014, Perret, Plush et al. 2017, Almberg, Halldin et al. 2018, Hall, Blackley et al. 2019, Go and Cohen 2020, Leonard, Zulfikar et al. 2020).

(Page 33, Lines 665-666) The limited number of studies meant that it was unclear whether the prevalence of pneumoconiosis had also changed over time in countries other than the USA.

11. Please check the forest plot if the study included in the meta-analysis shared the overlapping study period from the same data sources. That is not justice. 

Authors’ response: These have been checked and revised as required to ensure there are no overlapping study time periods. Studies included in meta-analysis have also been revised in response to Reviewer 4 #29-#33. 

The following section has been revised:

Results (Meta-analysis for CMDLD prevalence, Pages 24-25, Lines 484-488) Although two other studies reported results for CWP (CDC. 2003, Laney, Petsonk et al. 2012) and one study for PMF (Laney, Petsonk et al. 2012) prevalence, they were not included for those specific meta-analyses to avoid potential duplication with other studies. However, these three studies met the criteria for inclusion in the meta-analysis for the prevalence of other types of CMDLD. 

12. Conclusion section: Please concisely and briefly highlight in 3-5 points of your systematic review and meta-analysis. 

Authors’ response: The Conclusions section has been modified (Conclusions, Page 38):

Lines 787-795 This study summarizes published evidence from global studies on prevalence, mortality, and survival due to CMDLD, especially CWP. Overall summary pooled prevalence estimates highlighted that CWP remains a public health concern and were suggestive of an increase in prevalence among more recent time periods. CWP prevalence was also higher among underground coal miners than surface coal miners and varied by region within the USA. The international generalisability of findings was limited by most studies being sourced from the USA, while it was difficult to draw definitive conclusions from the small number of mortality and survival studies. Comparisons of findings across studies was difficult due to the inconsistent data collection and reporting methods used.

Lines 796-798 From a public health perspective, the continuing prevalence of CWP in the modern era highlight the need for ongoing surveillance and prevention efforts involving both dust monitoring and suppression in coal miners.

13. Please provide a table reporting in detail the results of your quality assessment, showing how each included study scored on every item of the Joanna Briggs quality assessment tool used.

Authors’ response: The following Table has been added to the Supplemental Materials:

S3 Table: Results of critical appraisal for studies included in systematic review

The following have been added to the manuscript (Results, Risk of Bias, Pages 16-17):

Lines 308-309 For 31 prevalence studies, the overall risk of bias score ranged from four to seven out of a total of nine domains in the JBI critical appraisal tool for prevalence studies (S3a Table)

Lines 323-324 Seven of eight mortality studies were considered to have a low risk of bias with scores of eight to nine out of 11 domains for JBI cohort tool (S3b Table). 

Lines 327-329 One survival study was classified as high risk of bias (score =5/11, S3b Table), having unclear information about follow-up, a non-representative study sample and ambiguous statistical analysis.

Reviewer #1:

14. The authors present "global" data on CWP based on metanalysis and pooling of data from multiple studies over many years (with most of the studies being US ones) The temporal changes described in the paper and critically important to understanding what is happening with CWP in the US are not apparent in any of the figures- which present data for each study that may cover over 40 years- during which the prevalence and outcomes of CWP may have changed dramatically. I think the authors need to determine a way to break the data into 5 or10 year intervals for presentation in the figures so that equals can be better compared to equals. Many of the primary papers cited have presented data in 5 or 10year intervals- so this should be possible.

Authors’ response: Additional meta-analyses have been carried out stratified by study time period, where sample sizes and number of included studies allowed.

Changes made to the manuscript are given below:

Abstract (Results, Lines 39-41) The pooled estimate of coal workers pneumoconiosis prevalence in the United States was higher in the 2000s than in the 1990s, consistent with published reports of increasing prevalence following decades of declining trends.

Abstract (Conclusion, Lines 45-47) The ongoing prevalence of occupational lung diseases among contemporary coal miners highlights the importance of respiratory surveillance and preventive efforts through effective dust control measures.

Methods (Statistical analysis, Page 10, Lines 206-210) Additional analysis by sub-groups such as coal mine type, country, geographical location (USA) and study time period were only carried out if there were at least two studies per strata. As such, the analyses stratified by time period (1990s compared with 2000s) were only possible for CWP and PMF in the USA, with the choice of these two periods determined by the included studies.

Methods (Statistical analysis, Page 10, Lines 211-212) Additional sub-group analyses by region were also conducted to generate summary estimates of CWP prevalence in the USA only in the most recent time period (2005-2009).

Results (Meta-analysis for CMDLD prevalence, By study time period, Page 27, Lines 537-545) Sub-group analysis by time period indicated that the prevalence of CWP in the USA was 3.0% (95% CI 2.8-3.2%) in the 1990s over two studies and 3.6% (95% CI 3.3-4.0%) in the 2000s over three studies (S5 Table). Tests for between-study heterogeneity found no evidence of heterogeneity for the 1990s (I2 =0.0%, P =0.901) and high, significant heterogeneity (I2 =82.5%, P =0.004) for the 2000s. For PMF, the pooled prevalence was 0.2% (95% CI 0.1-0.2%) across two studies in the 1990s with moderate but non-significant heterogeneity (I2 =63.5%, P =0.097) between studies (S5 Table). The corresponding estimate was 0.4% (95% CI 0.2-0.7%) across three studies during the 2000s with high and significant heterogeneity (I2 =95.4%, P <0.001).

S5 Table Forest plot from meta-analysis for prevalence of coal workers pneumoconiosis (top) and progressive massive fibrosis (bottom) in the United States by study time period (1990s versus 2000s). 

Coal workers pneumoconiosis

Progressive massive fibrosis 

Results (Meta-analysis for CMDLD prevalence, By study time period, Page 27, Lines 546-551) Finally, the pooled CWP prevalence among underground miners in the USA between 2005-2009 (two studies for each region) was 3.8% (95% CI 3.4-4.1%) nationally, 8.2% (95% CI 4.6-11.8%) among miners from Central Appalachia and 1.6% (95% CI 1.4-1.8%) for the rest of the country (S5 Fig). The between-study heterogeneity was moderate but non-significant for All USA (I2 =46.7%, P=0.171), high and significant for Central Appalachia (I2 =96.3%, P <0.001) and non-significant for ‘Rest of USA’ (I2 =0.0%, P=0.978) (S5 Fig).

S5 Fig Forest plot from meta-analysis for prevalence of coal worker’s pneumoconiosis (CWP) in the United States (USA) by region: All USA, Central Appalachia, and Rest of USA from 2005 to 2009. The term Central Appalachia refers to the region covered by states of Kentucky, Virginia, and West Virginia.

Discussion (Changes over time in CWP prevalence, Page 32, Lines 642-648): Sub-group analysis by time period suggested that the pooled prevalence of CWP in the USA was higher in the 2000s than during the 1990s, whereas although the summary point estimate for PMF was suggestive of a similar pattern, confidence intervals were overlapping. Moreover, the pooled prevalence of CWP was around 80% higher in the Central Appalachia region than the rest of the USA between 2005-2009. The limited number of studies, however, made it difficult to draw definitive conclusions on the magnitude and extent of any temporal changes in CWP, regardless of severity or PMF (the most severe form of CWP) prevalence.

Limitations (Page 37, Lines 762-767) The small number of studies also restricted exploration of the impact of factors such as study time period, regional location, mine characteristics including size or type and mining tenure on the pooled prevalence estimates in the USA itself. In particular, our ability to effectively summarize evidence for changes in prevalence of CWP over time was restricted by the wide heterogeneity in study time periods. This only allowed a small number of studies to be combined in sub-group analysis for any specific time period.

Please also see Responses to the Editor #10, Reviewer 2 #16, and Reviewer 4 #32

Reviewer 2:

15. This systematic review and meta-analysis aimed to evaluate the prevalence and the mortality due to Coal mine dust lung disease world wide. The authors achieved to give global prevalence data based on the available literature but were unable get an overall figure for mortality or survival due to the heterogeneity and non standardized way of estimation in the few studies that have addressed the question. As often in meta-analysis and in particular in this case, the conclusions are limited by the quality and representativeness of the original publications. The authors have made their best to extract meaningful and comparable results from the available studies, but were faced, among other hurdles, with a geographical bias with overrepresentation of studies from the USA. 

Authors’ response: Our thanks to the reviewer for these comments. The over-representation of studies from the USA is now shown visually in S1 Fig.

16. Another important message highlighted in the discussion is the increase of CWP prevalence over time, in particular from the 2000’s in the USA. This affirmation was mainly made on the quoted studies, however the data presented on Table 3 does not show striking higher prevalence for more recent period. For example, ref 30 that covers the period 2011-2014 has a low 0.2% prevalence whereas ref 25 shows a 3% prevalence for an earlier period. If the authors want to keep this as a major message of the present review, they should find a way to present this recent trend in a more convincing way.

Authors’ response: Due to different data sources used by the two references quoted above, this comparison is not completely valid. That is, reference 25 is based on Medicare data and reference 30 is based on surveillance data.

However, based on suggestions by Reviewer 1 #14 and Reviewer 4 #32, additional meta-analyses have been carried out that were stratified by time period (where sample sizes and the number of included studies were sufficiently large). 

A detailed discussion of changes made to the manuscript based on this revision have been given previously (see Reviewer 1 #14). Key results are also summarized below:

Results (Meta-analysis for CMDLD prevalence, By study time period, Page 27, Lines 537-545) Sub-group analysis by time period indicated that the prevalence of CWP in the USA was 3.0% (95% CI 2.8-3.2%) in the 1990s over two studies and 3.6% (95% CI 3.3-4.0%) in the 2000s over three studies (S5 Table). Tests for between-study heterogeneity found no evidence of heterogeneity for the 1990s (I2 =0.0%, P =0.901) and high, significant heterogeneity (I2 =82.5%, P =0.004) for the 2000s. For PMF, the pooled prevalence was 0.2% (95% CI 0.1-0.2%) across two studies in the 1990s with moderate but non-significant heterogeneity (I2 =63.5%, P =0.097) between studies (S5 Table). The corresponding estimate was 0.4% (95% CI 0.2-0.7%) across three studies during the 2000s with high and significant heterogeneity (I2 =95.4%, P <0.001).

Please also see Responses to the Editor #10, Reviewer 1 #14 and Reviewer 4 #32

17. Discussion should also be made on the possibility for better or earlier diagnosis in recent studies due to more systematic screening such as the use of CT scans.

Authors’ response: While it is true that better or earlier diagnosis through more systematic screening such as use of CT scans may lead to more cases of coal mine dust lung disease, all the reported prevalence estimates were based on cases diagnosed by radiographic X-rays only. As such, more recent estimates reported by included studies would not have been impacted by improved diagnostic techniques. 

The following has been added to the Discussion (Changes over time in CWP/PMF prevalence, Page 32, Lines 657-659):

Given that all included studies defined CWP cases based on the ILO diagnostic criteria (International Labour Organization 2011), reported prevalence estimates are unlikely to have been impacted by more sensitive diagnostic techniques such as high-resolution computed tomography screening (Go and Cohen 2020). 

18. I do not understand the forest plot of figure 7: Do the studies listed under “All USA” include or exclude data from central Appalachia.

Authors’ response: For Figure 7, the studies listed under “All USA” includes data from the whole country, including Central Appalachia. 

For additional clarity, the following has been added to the Results (Meta-analysis for CMDLD prevalence, Page 26, Lines 517-520):

By contrast, the pooled CWP prevalence nationally among underground miners in the USA (including Central Appalachia) across five studies was 3.7% (95% CI 3.0-4.5%), among Central Appalachian miners across two studies was 8.3% (95% CI 4.8-11.8% and among miners in the rest of the USA was 1.6% (95% CI 1.4-1.8%) (Fig 7). 

Forest plot has been revised also.

Fig 7a Forest plot from meta-analysis for prevalence of coal worker’s pneumoconiosis (CWP) in the United States (USA) by region: All USA, Central Appalachia, and Rest of USA. The red dashed line corresponds to the overall effect size to guide the eye. The term Central Appalachia refers to the region covered by states of Kentucky, Virginia, and West Virginia.

Please also responses to Reviewer 4 #28-#30.

19. Table 7 on mortality and survival is particularly frustrating to read. At least 5 different ways of evaluating mortality have been used making any comparison between studies a nightmare. Moreover, the type of CMDLD accounted for the cause of death varied between studies, some of them including COPD. I understand that due to this wide methodologic variability summary estimates could not be generated. However, I found a bit limited to conclude only with the statement that CWP is associated with an increased in mortality, which is common knowledge. I wonder if a crude case-fatality rate comparison would be possible and informative?

Authors’ response: Our thanks to the reviewer for recognising the difficulties in comparing the studies on mortality. As suggested, we looked at possibility of comparing crude case fatality rates. However, of the eight included papers only three (Attfield and Kuempel 2008, Miller and MacCalman 2010, Graber, Stayner et al. 2014) presented sufficient information to allow their calculation. 

The following has been added to the Manuscript (Results, Mortality, Page 28, Lines 557-559):

Crude fatality rate for pneumoconiosis among coal miners in the USA was estimated to be 4.3% (1969-1993) (Attfield and Kuempel 2008) and 4.6% (1969-2007) (Graber, Stayner et al. 2014). Corresponding estimate in the UK (1959-2005) was 1.6% (Miller and MacCalman 2010). 

20. Ref 17 is the only study assessing the prevalence of COPD in the miners. The data of 14.6% is about the double of the prevalence found in the adult population, but is similar to the prevalence reported for active smokers. Besides the exact definition of COPD applied to such survey, the confounding factor of smoking should of course be accounted for. Although COPD was not the main topic of this review a short discussion on these limitations would be useful.

Authors’ response: The following commentary has been added to the Discussion (Prevalence estimates for other CMDLD, Page 34, Lines 702-707):

The reported prevalence of COPD among underground miners in the Ukraine of 14.9% (Graber, Cohen et al. 2012), was about double that of the prevalence found in the adult population over the same time period but was similar to that reported for smokers (Halbert, Natoli et al. 2006). Given that smoking is the primary risk factor for COPD, these reported estimates are likely to have been impacted by confounding through smoking, especially given around 62% of the Ukraine study sample were current smokers.

Reviewer 3:

21. Coal workers’ pneumoconiosis (CWP), also known as black lung disease, is caused by long-term exposure to coal dust. Coal miners are vulnerable to CWP, as their work environment is highly exposed to coal dust. Many coal miners are likely to develop CWP due to an increase in coal production and utilization, especially in developing countries. CWP with other diseases including mixed dust pneumoconiosis, silicosis comprises the Coal mine dust lung disease (CMDLD). The authors of this manuscript went a great length in collecting and controlling the quality of data, which to reduce biases and to remove duplication, to perform a systematic review and meta-analysis of published research on prevalence, mortality, and survival for CMDLD. Although the objective might be a little broad, the objective is still clear. The pipeline of the authors research is well thought of and the meta-analysis is reasonable in which in the end forest plots were used to show each studies effect and significance was also clear.

Authors’ response: Our thanks to the reviewer for these comments

22. As the authors have mentioned biases still may exist in the results, however, is understandable as such biases are not limited to the authors research but a common concern in research conducting meta-analysis. Although such biases might still be a concern, the authors have well stated it in the limitation section of their manuscript and took account for it in interpreting their results. The authors interpretation of their results was interesting, and the logic was reasonable although there would need more follow up research to conclude in their findings. For that reason, the authors well positioned their interpretation in the discussion section in which the authors research may promote and contribute to future research, which is much needed in directing government policies for the health and welfare of coal miners.

Authors’ response: Our thanks to the reviewer for these comments

23. Suggestion to the study (please note the authors have no need to address or follow up on the suggestions): The manuscript would have been more enriched if the authors would have gone another depth in considering factors such as the age range of the minors of each study, as the age range of the minors might have changed during periods of time and different between locations in which older age might increase the risk of the disease. 

Authors’ response: Our ability to consider impact of age range of miners on the reported prevalence estimates was limited by the lack of information on the age range of miners in the study cohorts and/or the provision of prevalence estimates stratified by age range in the included studies. 

However, the following general comment has been added to the Discussion (Changes over time in CWP prevalence, Page 32, Lines 660-664):

The reported changes in prevalence of CWP and/or PMF in the USA may also reflect changes in the age range of coal miners over time and/or between geographical locations with older age increasing risk of disease. Our ability to consider the impact of age on reported prevalence estimates was limited by the lack of prevalence estimates stratified by age group across the included studies.

24. Although the authors consider other diseases of CMDLD, the authors mainly focus on CWP, which is reasonable, however, for the objective of the study other meta-analysis on related diseases such as Anthracosis would have provided a greater view. The authors do not have to address these suggestions as the limitation of data availability is understandable and the suggestions may be subjects for other studies and is not the focus of this manuscript.

Authors’ response: This is an interesting suggestion. However, as stated by the reviewer, it is outside scope of this manuscript and the current research questions for the systematic review/meta-analysis. 

No changes have been made to the manuscript.

25. One minor concern that the authors should consider and address: Due to the limitation of the data focusing on the studies conducted in the United States was not only reasonable but strategical and effective in addressing the objective of this study that may contribute to the national perspective on the current CWP burden in Australia and furthermore, guiding the Australian government policies for the health and welfare of coal miners. However, one minor concern would be as the studies used for the meta-analysis are mainly from the United States (n = 29) and the second is only followed by 4 studies from China, using the term ‘Global patterns’ in the title might be misleading and is a slight concern. To the best of my knowledge the patterns that are emphasized in the manuscript and the logical deduction the authors made are focused on studies conducted in the United States. Except for the fact that studies from 7 countries were used for the meta-analysis if there is no statistical evidence the pattern the authors emphasize is global, I would suggest the authors to consider rephrasing, which could be done by doing just one of the following.:1) Clarify or summarize what were the global patterns observed and then emphasize the patterns observed in the United states, as the patterns referring in the title and patterns emphasized in the discussion might be confused. 2) Consider rephrasing the title, and if the term ‘global’ is still preferred, it would be recommended to rephrase the title for example something like ‘A systematic review and meta-analysis on global studies of ~

Authors’ response: We have now rephrased the title to 

“A systematic review and meta-analysis on international studies of prevalence, mortality and survival due to coal mine dust lung disease”

Reviewer #4:

26. This is an important and well written manuscript reporting results from a well conducted meta-analysis of prevalence estimates for the diseases which encompass coal mine workers lung disease (CMWLD). It is the first systematic review and meta-analysis on the global prevalence of coal workers pneumoconiosis (CWP) and other CMWLD-spectrum disorders. These findings have important public health implications as they highlight the substantial burden of CMWLD and are suggestive that dust control standards have failed miners historically and need more stringent limits and enforcement. The strengths of this study include: that it followed the PRISMA guidelines; the thorough search for articles; the review and data abstraction processes which included robust quality control procedures; and, the assessment of potential bias using the Joanna Briggs Institute (JBI) critical appraisal tools. 

Authors’ response: Our thanks to the reviewer for these comments.

27. The main issue, which is addressable, is that prevalence estimates were combined despite significant study heterogeneity. The authors justify this by stating that all studies included in the meta-analysis used the standardized International Labour Organization (ILO) diagnostic criteria. While this is very important, an equally important criterium for combining prevalence estimates is that they be drawn from the same type of population, for example all from the general population or all from a population of coal miners. Combining estimates in the face of significant heterogeneity is ill advised. Three instances of these are given below

Authors’ response: The reported meta-analyses have been revised accordingly. Additonal commentary on general rules used for combining studies have been added: 

Methods (Statistical analysis, Pages 9-10, Lines 196-202) Only studies that reported both population size and number of disease cases and were deemed to be sufficiently homogenous in terms of study characteristics including geographical location and sampled population, were retained for the meta-analysis. This determination was subjective, however it meant for example that studies based on worker compensation data or other non-surveillance-based data sources were not combined with those based on population-based surveillance of coal miners. 

Results (Study selection, Page 11, Lines 239-242) Of the remaining 119 full-text articles that were evaluated for eligibility, 40 were retained for the review (31 prevalence, 8 mortality and 1 survival), of which 15 were eligible for meta-analyses (15 prevalence, 0 mortality and 0 survival).

Results (Meta-analysis for CMDLD prevalence, Page 24, Lines 479-484) Fifteen (48%) of 31 prevalence studies were retained for the meta-analysis. Excluded studies typically did not report overall estimates (n=7) (CDC. 2006, CDC. 2007, Laney and Attfield 2014, Reynolds, Blackley et al. 2017, Blackley, Halldin et al. 2018, Blackley, Reynolds et al. 2018, Kurth, Laney et al. 2020), or did not present sufficient information to be included (n=3) (Naidoo, Robins et al. 2004, Tor, Ozturk et al. 2010, Vallyathan, Landsittel et al. 2011). Three studies based on compensation databases (Graber, Harris et al. 2017, Almberg, Halldin et al. 2018, Arif, Paul et al. 2020) were also dropped as denominator was not comparable to surveillance-based studies. In addition, the single study from the UK (Scarisbrick and Quinlan 2002) and one study each that reported estimates for only COPD (Graber, Cohen et al. 2012) or silicosis (CDC. 2000) were excluded.

Discussion (Summary of main findings, Page 30, Lines 597-601) Thirty-one articles on CMDLD prevalence were included in this review of which 15 were retained for meta-analysis. Excluded articles typically either reported estimates for population sub-groups only, did not present sufficient information to be included or could not be combined with other studies due to wide variability in data sources.

Discussion (Summary of main findings, Page 30, Lines 603-605) This also meant that 12 of 15 studies included in the meta-analysis were from the USA limiting our ability to understand the contemporary burden of CMDLD in other coal-producing countries.

Discussion (Summary of main findings, Pages 30-31, Lines 617-620) Moreover, prevalence estimates from the USA were only combined if they were drawn from the same type of population. In practise this meant that studies based on compensation databases for example were excluded as denominator was not comparable to population-based surveillance databases of coal miners. 

Please also see responses to Reviewer 4 #28-31

28. There appears to be significant heterogeneity among the studies included in the analysis of overall prevalence of coal worker’s pneumoconiosis (CWP) among underground coal miners. (Figure 2) - the source may be combining studies from different regions of the world so the degree to which this meta-prevalence is generalizable to a ‘global population’ is unclear.

Authors’ response: Given the degree of heterogeneity among the studies we have modified the meta-analysis for overall prevalence of coal worker’s pneumoconiosis (CWP) among underground coal miners. (Figure 2) by only including relevant studies from the USA based on surveillance methodologies and three studies from China and doing the analyses stratified by country (USA versus China).

The following changes have been made to the manuscript:

Title As per Reviewer3 (Comment #25) the Title has been revised to “A systematic review and meta-analysis on international studies of prevalence, mortality and survival due to coal mine dust lung disease”.

Abstract (Results, Lines 36-39) Of the prevalence estimates, fifteen (12 from the United States) were retained for the meta-analysis. The overall pooled prevalence estimate for coal workers pneumoconiosis among underground miners was 3.7% (95% CI 3.0-4.5%) with high heterogeneity between studies.

Methods (Statistical analysis, Pages 9-10, Lines 196-202) Only studies that reported both population size and number of disease cases and were deemed to be sufficiently homogenous in terms of study characteristics including geographical location and sampled population, were retained for the meta-analysis. This determination was subjective, however it meant for example that studies based on worker compensation data or other non-surveillance-based data sources were not combined with those based on population-based surveillance of coal miners. 

Methods (Statistical analysis, Page 10, Lines 206-208) Additional analysis by sub-groups such as coal mine type, country, geographical location (USA) and study time period were only carried out if there were at least two studies per strata.

Results (Meta-analysis for CMDLD prevalence, Page 24, Lines 479-484) Fifteen (48%) of 31 prevalence studies were retained for the meta-analysis. Excluded studies typically did not report overall estimates (n=7) (CDC. 2006, CDC. 2007, Laney and Attfield 2014, Reynolds, Blackley et al. 2017, Blackley, Halldin et al. 2018, Blackley, Reynolds et al. 2018, Kurth, Laney et al. 2020), or did not present sufficient information to be included (n=3) (Naidoo, Robins et al. 2004, Tor, Ozturk et al. 2010, Vallyathan, Landsittel et al. 2011). Three studies based on compensation databases (Graber, Harris et al. 2017, Almberg, Halldin et al. 2018, Arif, Paul et al. 2020) were also dropped as denominator was not comparable to surveillance-based studies. In addition, the single study from the UK (Scarisbrick and Quinlan 2002) and one study each that reported estimates for only COPD (Graber, Cohen et al. 2012) or silicosis (CDC. 2000) were excluded.

Results (Meta-analysis for CMDLD prevalence, Page 25, Lines 489-494) Overall pooled estimate for CWP prevalence (Fig 2) over eight studies was 3.7% (95% CI 2.9-4.5%) with a high and significant heterogeneity between studies (I2 =98.9%, P <0.001). Sub-group analysis by country indicated that summary point estimates were very similar for USA (3.7% (95% CI 3.0-4.5%)) over five studies and China (3.6% (95% CI 1.6-5.6%)) over three studies, although confidence intervals were wide. Sensitivity analysis indicated that these estimates were not sensitive to individual studies (S3 Fig).

Fig 2 Forest plot from meta-analysis for overall prevalence of coal worker’s pneumoconiosis (CWP) among underground coal miners by country. Abbreviations are USA United States. The red dashed line corresponds to the overall effect size to guide the eye.

S3 Fig Sensitivity analysis for the influence of individual studies on the pooled prevalence estimates from meta-analysis for overall prevalence of coal workers pneumoconiosis among underground coal miners.

Results (Meta-analysis for CMDLD prevalence, Page 26, Lines 515-524) Summary estimates of CWP prevalence among surface miners in the USA across three studies was 1.8% (95% CI 1.6-2.0%) with low and non-significant heterogeneity (I2 =24.8%, P =0.248) (Fig 6) between studies. By contrast, the pooled CWP prevalence nationally among underground miners in the USA (including Central Appalachia) across five studies was 3.7% (95% CI 3.0-4.5%), among Central Appalachian miners across two studies was 8.3% (95% CI 4.8-11.8% and among miners in the rest of the USA was 1.6% (95% CI 1.4-1.8%) (Fig 7). There was high and significant heterogeneity between studies for All USA (I2 =98.3%, P <0.001) and regional subgroup of Central Appalachia (I2 =95.3%, P <0.001) although estimates were robust to influence of individual studies (S4 Table). No evidence of between study heterogeneity was found for ‘Rest of USA’ (I2 =0.0%, P=0.978). 

Fig 6 Forest plot from meta-analysis for overall prevalence of coal worker’s pneumoconiosis (CWP) among surface coal miners in the United States (USA). The red dashed line corresponds to the overall effect size to guide the eye.

S4 Table Sensitivity analyses for the influence of individual studies on pooled prevalence estimates for coal mine lung disease in the United States. Included meta-analyses are for coal workers pneumoconiosis (CWP) by disease severity (progressive massive fibrosis, advanced coal workers pneumoconiosis), mine type (surface), region (Central Appalachia versus overall United States) and r-type opacities. Each plot shows the calculated prevalence estimates with the named study omitted.

Progressive massive fibrosis

Advanced CWP Surface mines-CWP

r-type opacities CWP (Central Appalachia, includes states Kentucky, Virginia, and West Virginia in the USA)

CWP (National, all USA)

Discussion (Summary of main findings, Page 30, Lines 603-605) This also meant that 12 of 15 studies included in the meta-analysis were from the USA limiting our ability to understand the contemporary burden of CMDLD in other coal-producing countries.

Discussion (Changes over time in CWP/PMF prevalence, Page 33, Lines 670-671) Moreover, sub-group analysis by country indicated that the pooled point prevalence estimates of CWP in USA and China were very similar.

Discussion (Mine characteristics and CWP prevalence, Page 33, Lines 685-686) Although the summary estimate of CWP prevalence among surface miners was around 51% lower than for underground miners in the USA (1.8 versus 3.7%), it was not negligible.

Limitations (Pages 36-37, Lines 751-753) Of 15 studies retained for calculation of pooled estimates, 12 were from the USA and three from China. 

Limitations (Page 37, Line 761-767) A lack of studies from countries besides the USA limited our ability to make comparisons between countries. The small number of studies also restricted exploration of the impact of factors such as study time period, regional location, mine characteristics including size or type and mining tenure on the pooled prevalence estimates in the USA itself. 

Forest plots have been updated and relevant Tables have also been amended accordingly 

Please also see Responses below for comments Reviewer 4 #29 to #30 that also address the inclusion/exclusion of studies from meta-analysis.

29. The study by Arif AA et. al. (2020) estimated population rates (county-level) of CWP while the other studies in the group it is combined with used surveillance methodologies, so their denominator is coal miners. These should not be in the same meta-analysis.

Authors’ response: The study by Arif et al (2020) has been excluded from the meta-analysis for CWP prevalence.

The following changes have been made to the manuscript 

Abstract (Results, Lines 36-39) Of the prevalence estimates, fifteen (12 from the United States) were retained for the meta-analysis. The overall pooled prevalence estimate for coal workers pneumoconiosis among underground miners was 3.7% (95% CI 3.0-4.5%) with high heterogeneity between studies.

Methods (Statistical analysis, Pages 9-10, Lines 199-202) This determination was subjective, however it meant for example that studies based on worker compensation data or other non-surveillance-based data sources were not combined with those based on population-based surveillance of coal miners.

Results (Meta-analysis for CMDLD prevalence, Page 24, Line 479) Fifteen (48%) of 31 prevalence studies were retained for the meta-analysis. 

Results (Meta-analysis for CMDLD prevalence, Page 24, Lines 481-483) Three studies based on compensation databases (Graber, Harris et al. 2017, Almberg, Halldin et al. 2018, Arif, Paul et al. 2020) were also dropped as denominator was not comparable to surveillance-based studies.

Results (Meta-analysis for CMDLD prevalence, Page 25, Lines 489-494) Overall pooled estimate for CWP prevalence (Fig 2) over eight studies was 3.7% (95% CI 2.9-4.5%) with a high and significant heterogeneity between studies (I2 =98.9%, P <0.001). Sub-group analysis by country indicated that summary point estimates were very similar for USA (3.7% (95% CI 3.0-4.5%)) over five studies and China (3.6% (95% CI 1.6-5.6%)) over three studies, although confidence intervals were wide. Sensitivity analysis indicated that these estimates were not sensitive to individual studies (S3 Fig).

Results (Meta-analysis for CMDLD prevalence, Page 26, Lines 517-524) By contrast, the pooled CWP prevalence nationally among underground miners in the USA (including Central Appalachia) across five studies was 3.7% (95% CI 3.0-4.5%), among Central Appalachian miners across two studies was 8.3% (95% CI 4.8-11.8% and among miners in the rest of the USA was 1.6% (95% CI 1.4-1.8%) (Fig 7). There was high and significant heterogeneity between studies for All USA (I2 =98.3%, P <0.001) and regional subgroup of Central Appalachia (I2 =95.3%, P <0.001) although estimates were robust to influence of individual studies (S4 Table). No evidence of between study heterogeneity was found for ‘Rest of USA’ (I2 =0.0%, P=0.98). 

Discussion (Changes over time in CWP/PMF prevalence, Page 33, Lines 670-671) Moreover, sub-group analysis by country indicated that the pooled point prevalence estimates of CWP in USA and China were very similar.

Discussion (Regional variation in CMDLD prevalence (USA only), Page 34, Lines 693-695) These regional patterns may be reflected in the summary pooled estimates for CWP among underground miners in the USA which were significantly higher for Central Appalachia than the ‘Rest of USA’ or nationally.

Limitations (Pages 36-37, Line 751-753) Of 15 studies retained for calculation of pooled estimates, 12 were from the USA and three from China. 

Forest plots have been updated and relevant Tables have also been amended accordingly 

30. The authors note that the prevalence estimates from the studies using workers compensation program data and those using surveillance data likely are from very different populations. The population included the workers compensation program are likely skewed toward those with longer mining tenure and disease symptoms, and so likely overestimate the prevalence of CMDLD. In the forest plots in Fig 3 those using workers compensation program data (Graber et al, 2017 and Almberg et al., 2018, respectively) are clear outliers. As such they should not be combined with the other studies in the group that are more population -based surveillance methodologies. This is further supported by the sensitivity analyses which “indicated that pooled estimates were particularly sensitive to the study by Graber”)

Authors’ response: Our thanks to the reviewer for pointing this out. We have now dropped the two studies that used workers compensation program data (Graber et al, 2017 and Almberg et al., 2018, respectively) from the meta-analysis for prevalence of progressive massive fibrosis in the USA (Figure 3). The study by Graber et al (Graber, Harris et al. 2017) was also dropped from the meta-analysis for advanced CWP, (Figure 4).

The following changes have been made to the manuscript: 

Abstract (Results, Lines 36-37) Of the prevalence estimates, fifteen (12 from the United States) were retained for the meta-analysis.

Methods (Statistical analysis, Pages 9-10, Lines 199-202) This determination was subjective, however it meant for example that studies based on worker compensation data or other non-surveillance-based data sources were not combined with those based on population-based surveillance of coal miners.

Results (Meta-analysis for CMDLD prevalence, Page 24, Lines 481-483) Three studies based on compensation databases (Graber, Harris et al. 2017, Almberg, Halldin et al. 2018, Arif, Paul et al. 2020) were also dropped as denominator was not comparable to surveillance-based studies.

Results (Meta-analysis for CMDLD prevalence, Page 25, Lines 498-501) Analyses of CWP subtypes showed that the prevalence estimates of PMF in the USA was 0.3% (95% CI 0.2-0.5%) over five studies (Fig 3) and 0.9% (95% CI 0.6-1.2%) for advanced CWP over three studies (Fig 4). For both these diseases, inter-study heterogeneity was high (PMF: I2 =97.8%, advanced CWP I2 =94.8%) and statistically significant (P <0.001).

Fig 3 Forest plot from meta-analysis for overall prevalence of progressive massive fibrosis (PMF) in the United States (USA). The red dashed line corresponds to the overall effect size to guide the eye.

Fig 4 Forest plot from meta-analysis for overall prevalence of advanced coal workers pneumoconiosis in the United States (USA).

Discussion (Summary of main findings, Page30, Lines 606-609) Combining the study-specific estimates through meta-analysis indicated that the pooled prevalence of CWP among underground miners was 3.7%, although confidence intervals were wide. Corresponding estimates for PMF, advanced CWP and r-type opacities in the USA were 0.3%, 0.9% and 0.7% respectively.

Forest plots have been updated and relevant Tables have also been amended accordingly. 

31. The Graber et al 2014 study (1969/71- 2007) is an update of Attfield & Kuempel 2008 (1969/71- 1993) and the same outcomes were included in the metanalysis (pneumoconiosis (CWP, silicosis, COPD) so the reason for inclusion of both is unclear.

Authors’ response: Yes, the Graber 2014 study is an update of the Atfield & Kuempel 2008 study, however no meta-analysis was performed using results from either of these studies as these were both mortality studies. Both studies have only been included in the systematic review.

No changes have been made to the manuscript.

32. Given that the prevalence of CMWLD has changed significantly over the period of studies covered (1959 and 2019), the authors should consider stratifying the analyses by time if sample size permits.

Authors’ response: Additional meta-analyses have been carried out stratified by time, where sample sizes and number of included studies allowed.

The changes made to the manuscript have been described in full detail in Responses to Reviewer 1 #14.

Abstract (Results, Lines 39-41) The pooled estimate of coal workers pneumoconiosis prevalence in the United States was higher in the 2000s than in the 1990s, consistent with published reports of increasing prevalence following decades of declining trends.

Abstract (Conclusion, Lines 45-47) The ongoing prevalence of occupational lung diseases among contemporary coal miners highlights the importance of respiratory surveillance and preventive efforts through effective dust control measures.

Methods (Statistical analysis, Page 10, Lines 206-210) Additional analysis by sub-groups such as coal mine type, country, geographical location (USA) and study time period were only carried out if there were at least two studies per strata. As such, the analyses stratified by time period (1990s compared with 2000s) were only possible for CWP and PMF in the USA, with the choice of these two periods determined by the included studies.

Methods (Statistical analysis, Page 10, Lines 211-212) Additional sub-group analyses by region were also conducted to generate summary estimates of CWP prevalence in the USA only in the most recent time period (2005-2009).

Results (Meta-analysis for CMDLD prevalence, By study time period, Page 27, Lines 537-545) Sub-group analysis by time period indicated that the prevalence of CWP in the USA was 3.0% (95% CI 2.8-3.2%) in the 1990s over two studies and 3.6% (95% CI 3.3-4.0%) in the 2000s over three studies (S5 Table). Tests for between-study heterogeneity found no evidence of heterogeneity for the 1990s (I2 =0.0%, P =0.901) and high, significant heterogeneity (I2 =82.5%, P =0.004) for the 2000s. For PMF, the pooled prevalence was 0.2% (95% CI 0.1-0.2%) across two studies in the 1990s with moderate but non-significant heterogeneity (I2 =63.5%, P =0.097) between studies (S5 Table). The corresponding estimate was 0.4% (95% CI 0.2-0.7%) across three studies during the 2000s with high and significant heterogeneity (I2 =95.4%, P <0.001).

Results (Meta-analysis for CMDLD prevalence, By study time period, Page 27, Lines 546-551) Finally, the pooled CWP prevalence among underground miners in the USA between 2005-2009 (two studies for each region) was 3.8% (95% CI 3.4-4.1%) nationally, 8.2% (95% CI 4.6-11.8%) among miners from Central Appalachia and 1.6% (95% CI 1.4-1.8%) for the rest of the country (S5 Fig). The between-study heterogeneity was moderate but non-significant for All USA (I2 =46.7%, P=0.171), high and significant for Central Appalachia (I2 =96.3%, P <0.001) and non-significant for ‘Rest of USA’ (I2 =0.0%, P=0.978) (S5 Fig).

Discussion (Changes over time in CWP prevalence, Page 32, Lines 642-648): Sub-group analysis by time period suggested that the pooled prevalence of CWP in the USA was higher in the 2000s than during the 1990s, whereas although the summary point estimate for PMF was suggestive of a similar pattern, confidence intervals were overlapping. Moreover, the pooled prevalence of CWP was around 80% higher in the Central Appalachia region than the rest of the USA between 2005-2009. The limited number of studies, however, made it difficult to draw definitive conclusions on the magnitude and extent of any temporal changes in CWP, regardless of severity or PMF (the most severe form of CWP) prevalence.

Limitations (Page 37, Lines 762-767) The small number of studies also restricted exploration of the impact of factors such as study time period, regional location, mine characteristics including size or type and mining tenure on the pooled prevalence estimates in the USA itself. In particular, our ability to effectively summarize evidence for changes in prevalence of CWP over time was restricted by the wide heterogeneity in study time periods. This only allowed a small number of studies to be combined in sub-group analysis for any specific time period.

Please also see Responses to the Editor #10, Reviewer 1 #14, and Reviewer 2 #16. 

Updated Forest Plots are shown in Responses to Reviewer 1#14.

33. The authors do an impressive bias assessment using the JBI tool and present robust results. Curiously, they did use this information to inform the analysis. Sensitivity analyses based on the findings would be informative.

Authors’ response: Additional sensitivity analyses were performed to explore whether the pooled prevalence estimates for CWP among underground coal miners in the USA were impacted by the quality (risk of bias) of the five included studies. Further differentiation by study quality was not possible for PMF, advanced CWP or r-type opacities due to the limited number of studies. 

The following changes have been made to the manuscript:

Methods (Statistical analysis, Page 10, Lines 213-216) Sensitivity analyses were carried out to explore the impact of retaining only the four high quality (low risk of bias) studies on the pooled prevalence estimates for CWP among underground coal miners in the USA. Sub-group analysis by risk-of-bias for other disease types was not possible due to the limited number of studies. 

Results (Page 27, Lines 533-535) Sensitivity analysis by study quality (S4 Fig) gave a pooled CWP prevalence of 3.7% (95% CI 2.8-4.6%) over four high quality (low risk of bias) studies, with the point estimate being similar to the original analysis which did not exclude studies on basis of their quality. 

S4 Fig Sensitivity analysis for the influence of study quality (risk of bias) on the pooled prevalence estimates for coal workers pneumoconiosis among underground coal miners in the United States. 

Discussion (Summary of main findings, Page 31, Lines 621-622) We found no evidence of publication bias or bias related to study quality (as measured by risk of bias).

An additional Supplemental Figure has been added (S4 Fig).

34. Beginning line 70 “The increase in CWP prevalence has occurred even among miners who were subject to regulatory guidelines their entire working life. Consider whether citation #15 is also appropriate here.

Authors’ response: This reference has now been added (Introduction, Page 4, Line 73):

The increase in CWP prevalence has occurred even among miners who were subject to regulatory guidelines their entire working life (Graber, Harris et al. 2017).

35. Beginning line 151: please clarify that these are states with the United States

Authors’ response: For clarity we have modified this sentence (Methods, Data extraction, Page 7, Lines 153-155): 

The region covered by three states within the USA, Kentucky, Virginia, and West Virginia, is referred to throughout this review as ‘Central Appalachia’ 

36. Line 116 (and Line 28) , Web of Science is included as a database but is not included in Table S1

Authors’ response: No, there was no specific search strategy for Web of Science. This database was only used to look for other references that had cited studies in the systematic review which were identified on the basis of their “Author and Title”. Hence, no specific search query was generated for this database to be included in Table S1.

No changes have been made to the manuscript.

37. Beginning line 194: Please clarify how homogeneity of study characteristics was assessed.

Authors’ response: This was initially done on a subjective basis with no formal assessment. However, in the revised analysis, we also looked at the impact of excluding/including specific studies on the I2 measure of between-study heterogeneity.

For clarity the following has been added:

Methods (Statistical analysis, Pages 9-10, Lines 196-202) Only studies that reported both population size and number of disease cases and were deemed to be sufficiently homogenous in terms of study characteristics including geographical location and sampled population, were retained for the meta-analysis. This determination was subjective, however it meant for example that studies based on worker compensation data or other non-surveillance-based data sources were not combined with those based on population-based surveillance of coal miners. 

Methods (Statistical analysis, Pages 10-11, Lines 220-223) Heterogeneity between study-specific estimates was assessed with the Q and I2 statistics (Higgins, Thompson et al. 2003) with I2 values of 25%, 50% and 75% being the cut-offs for indicating low, moderate and high heterogeneity respectively. In the initial exploratory meta-analysis, we looked at the impact of excluding/including specific studies on the I2 measure of between-study heterogeneity. 

Please see also responses to Reviewer 4 #28 -#30. 

References

Almberg, K. S., C. N. Halldin, D. J. Blackley, A. S. Laney, E. Storey, C. S. Rose, L. H. T. Go and R. A. Cohen (2018). "Progressive Massive Fibrosis Resurgence Identified in U.S. Coal Miners Filing for Black Lung Benefits, 1970-2016." Ann Am Thorac Soc 15(12): 1420-1426.

Arif, A. A., R. Paul, E. Delmelle, C. Owusu and O. Adeyemi (2020). "Estimating the prevalence and spatial clusters of coal workers' pneumoconiosis cases using medicare claims data, 2011-2014." Am J Ind Med 63(6): 478-483.

Attfield, M. D. and E. D. Kuempel (2008). "Mortality among U.S. underground coal miners: a 23-year follow-up." Am J Ind Med 51(4): 231-245.

Blackley, D. J., C. N. Halldin and A. S. Laney (2018). "Continued Increase in Prevalence of Coal Workers' Pneumoconiosis in the United States, 1970-2017." Am J Public Health 108(9): 1220-1222.

Blackley, D. J., C. N. Halldin, M. L. Wang and A. S. Laney (2014). "Small mine size is associated with lung function abnormality and pneumoconiosis among underground coal miners in Kentucky, Virginia and West Virginia." Occup Environ Med 71(10): 690-694.

Blackley, D. J., L. E. Reynolds, C. Short, R. Carson, E. Storey, C. N. Halldin and A. S. Laney (2018). "Progressive Massive Fibrosis in Coal Miners From 3 Clinics in Virginia." Jama 319(5): 500-501.

CDC. (2000). "Silicosis screening in surface coal miners--Pennsylvania, 1996-1997." MMWR 49(27): 612-615.

CDC. (2003). "Pneumoconiosis prevalence among working coal miners examined in federal chest radiograph surveillance programs--United States, 1996-2002." MMWR 52(15): 336-340.

CDC. (2006). "Advanced cases of coal workers' pneumoconiosis--two counties, Virginia, 2006." MMWR 55(33): 909-913.

CDC. (2007). "Advanced pneumoconiosis among working underground coal miners--Eastern Kentucky and Southwestern Virginia, 2006." MMWR 56(26): 652-655.

Go, L. H. T. and R. A. Cohen (2020). "Coal Workers' Pneumoconiosis and Other Mining-Related Lung Disease: New Manifestations of Illness in an Age-Old Occupation." Clin Chest Med 41(4): 687-696.

Graber, J. M., R. A. Cohen, A. Basanets, L. T. Stayner, Y. Kundiev, L. Conroy, V. V. Mukhin, O. Lysenko, A. Zvinchuk and D. O. Hryhorczuk (2012). "Results from a Ukrainian-US collaborative study: prevalence and predictors of respiratory symptoms among Ukrainian coal miners." Am J Ind Med 55(12): 1099-1109.

Graber, J. M., G. Harris, K. S. Almberg, C. S. Rose, E. L. Petsonk and R. A. Cohen (2017). "Increasing Severity of Pneumoconiosis Among Younger Former US Coal Miners Working Exclusively Under Modern Dust-Control Regulations." J Occup Environ Med 59(6): e105-e111.

Graber, J. M., L. T. Stayner, R. A. Cohen, L. M. Conroy and M. D. Attfield (2014). "Respiratory disease mortality among US coal miners; results after 37 years of follow-up." Occup Environ Med 71(1): 30-39.

Halbert, R. J., J. L. Natoli, A. Gano, E. Badamgarav, A. S. Buist and D. M. Mannino (2006). "Global burden of COPD: systematic review and meta-analysis." Eur Respir J 28(3): 523-532.

Hall, N. B., D. J. Blackley, C. N. Halldin and A. S. Laney (2019). "Current Review of Pneumoconiosis Among US Coal Miners." Curr Environ Health Rep 6(3): 137-147.

Higgins, J. P., S. G. Thompson, J. J. Deeks and D. G. Altman (2003). "Measuring inconsistency in meta-analyses." BMJ 327(7414): 557-560.

International Labour Organization. (2011). "Guidelines for the use of the ILO International Classification of Radiographs of Pneumoconioses, Revised edition 2011. ." Retrieved 14/01/2021, from http://www.ilo.org/safework/info/publications/WCMS_168260/lang--en/index.htm.

Kurth, L., A. S. Laney, D. J. Blackley and C. N. Halldin (2020). "Prevalence of spirometry-defined airflow obstruction in never-smoking working US coal miners by pneumoconiosis status." Occupational and Environmental Medicine 77(4): 265-267.

Laney, A. S. and M. D. Attfield (2010). "Coal workers' pneumoconiosis and progressive massive fibrosis are increasingly more prevalent among workers in small underground coal mines in the United States." Occup Environ Med 67(6): 428-431.

Laney, A. S. and M. D. Attfield (2014). "Examination of potential sources of bias in the US Coal Workers' Health Surveillance Program." Am J Public Health 104(1): 165-170.

Laney, A. S., D. J. Blackley and C. N. Halldin (2017). "Radiographic disease progression in contemporary US coal miners with progressive massive fibrosis." Occup Environ Med 74(7): 517-520.

Laney, A. S., E. L. Petsonk, J. M. Hale, A. L. Wolfe and M. D. Attfield (2012). "Potential determinants of coal workers' pneumoconiosis, advanced pneumoconiosis, and progressive massive fibrosis among underground coal miners in the United States, 2005-2009." Am J Public Health 102 Suppl 2: S279-283.

Leonard, R., R. Zulfikar and R. Stansbury (2020). "Coal mining and lung disease in the 21st century." Current Opinion in Pulmonary Medicine 26(2): 135-141.

Miller, B. G. and L. MacCalman (2010). "Cause-specific mortality in British coal workers and exposure to respirable dust and quartz." Occup Environ Med 67(4): 270-276.

Naidoo, R. N., T. G. Robins, A. Solomon, N. White and A. Franzblau (2004). "Radiographic outcomes among South African coal miners." Int Arch Occup Environ Health 77(7): 471-481.

Perret, J. L., B. Plush, P. Lachapelle, T. S. Hinks, C. Walter, P. Clarke, L. Irving, P. Brady, S. C. Dharmage and A. Stewart (2017). "Coal mine dust lung disease in the modern era." Respirology 22(4): 662-670.

Reynolds, L. E., D. J. Blackley, A. S. Laney and C. N. Halldin (2017). "Respiratory morbidity among U.S. coal miners in states outside of central Appalachia." Am J Ind Med 60(6): 513-517.

Scarisbrick, D. and R. Quinlan (2002). "Health surveillance for coal workers' pneumoconiosis in the United Kingdom 1998–2000." Annals of Occupational Hygiene 46(suppl_1): 254-256.

Suarthana, E., A. S. Laney, E. Storey, J. M. Hale and M. D. Attfield (2011). "Coal workers' pneumoconiosis in the United States: regional differences 40 years after implementation of the 1969 Federal Coal Mine Health and Safety Act." Occup Environ Med 68(12): 908-913.

Tor, M., M. Ozturk, R. Altin and A. H. Cimrin (2010). "Working conditions and pneumoconiosis in Turkish coal miners between 1985 and 2004: a report from Zonguldak coal basin, Turkey." Tuberk Toraks 58(3): 252-260.

Vallyathan, V., D. P. Landsittel, E. L. Petsonk, J. Kahn, J. E. Parker, K. T. Osiowy and F. H. Green (2011). "The influence of dust standards on the prevalence and severity of coal worker's pneumoconiosis at autopsy in the United States of America." Arch Pathol Lab Med 135(12): 1550-1556.

---

## [Decision Letter · Decision Letter 1]

21 Jul 2021

A systematic review and meta-analysis on international studies of prevalence, mortality and survival due to coal mine dust lung disease

PONE-D-21-09488R1

Dear Dr. Baade,

We’re pleased to inform you that your manuscript has been judged scientifically suitable for publication and will be formally accepted for publication once it meets all outstanding technical requirements.

Kind regards,

Yuan-Pin Hsu

Academic Editor

PLOS ONE

Additional Editor Comments (optional):

Reviewers' comments:

Reviewer's Responses to Questions

**Comments to the Author**

1. If the authors have adequately addressed your comments raised in a previous round of review and you feel that this manuscript is now acceptable for publication, you may indicate that here to bypass the “Comments to the Author” section, enter your conflict of interest statement in the “Confidential to Editor” section, and submit your "Accept" recommendation.

Reviewer #1: All comments have been addressed

Reviewer #2: All comments have been addressed

Reviewer #3: All comments have been addressed

2. Is the manuscript technically sound, and do the data support the conclusions?

Reviewer #1: Yes

Reviewer #2: Yes

Reviewer #3: Partly

3. Has the statistical analysis been performed appropriately and rigorously? 

Reviewer #1: Yes

Reviewer #2: Yes

Reviewer #3: Yes

4. Have the authors made all data underlying the findings in their manuscript fully available?

Reviewer #1: Yes

Reviewer #2: Yes

Reviewer #3: Yes

5. Is the manuscript presented in an intelligible fashion and written in standard English?

Reviewer #1: Yes

Reviewer #2: Yes

Reviewer #3: Yes

6. Review Comments to the Author

Reviewer #1: I thought the authors did a nice job in responding to my concerns.

Reviewer #2: Substantial modification to the analysis and to every chapter of the manuscript have been made, in response to the reviewers and editor comments and queries. I find that this paper has gained in accuracy and intelligibility comparing to the initial version. The limitations due to the quality and heterogeneity of the starting material is now more precisely explained.

Reviewer #3: The authors of the paper have addressed my comments and I have no further comments on the manuscript.

7. PLOS authors have the option to publish the peer review history of their article (what does this mean?). If published, this will include your full peer review and any attached files.

Reviewer #1: **Yes: **David Mannino

Reviewer #2: No

Reviewer #3: **Yes: **Seung Han Baek

---

## [Editor Report · Acceptance letter]

26 Jul 2021

PONE-D-21-09488R1 

A systematic review and meta-analysis on international studies of prevalence, mortality and survival due to coal mine dust lung disease 

Dear Dr. Baade:

I'm pleased to inform you that your manuscript has been deemed suitable for publication in PLOS ONE. Congratulations! Your manuscript is now with our production department. 

Kind regards, 

on behalf of

Dr. Yuan-Pin Hsu 

Academic Editor

PLOS ONE